# KNOW YOUR ACTION SET: LEARNING ACTION RELATIONS FOR REINFORCEMENT LEARNING

**Ayush Jain**[*1]     **Norio Kosaka**[*2]     **Kyung-Min Kim**[2 4]     **Joseph J. Lim**[3†4‡]
[1]University of Southern California (USC), [2]NAVER CLOVA,
[3]Korea Advanced Institute of Science and Technology (KAIST), [4]NAVER AI Lab

## ABSTRACT

Intelligent agents can solve tasks in various ways depending on their available set of actions. However, conventional reinforcement learning (RL) assumes a fixed action set. This work asserts that tasks with varying action sets require reasoning of the relations between the available actions. For instance, taking a nail-action in a repair task is meaningful only if a hammer-action is also available. To learn and utilize such action relations, we propose a novel policy architecture consisting of a graph attention network over the available actions. We show that our model makes informed action decisions by correctly attending to other related actions in both value-based and policy-based RL. Consequently, it outperforms non-relational architectures on applications where the action space often varies, such as recommender systems and physical reasoning with tools and skills. [1]

## 1 INTRODUCTION

Imagine you want to hang a picture on the wall. You may start by placing a nail on the wall and then use a hammer to nail it in. However, if you do not have access to a hammer, you would not use the nail. Instead, you would try alternative approaches such as using a hook and adhesive-strips or a screw and a drill (Figure 1). In general, we solve tasks by choosing actions that interact with each other to achieve a desired outcome in the environment. Therefore, the best action decision depends not only on the environment but also on what other actions are available to use.

This work addresses the setting of varying action space in sequential decision-making. Typically reinforcement learning (RL) assumes a fixed action space, but recent work has explored variations in action space when adding or removing actions (Boutilier et al., 2018; Chandak et al., 2020a;b) or for generalization to unseen actions (Jain et al., 2020). These assume that the given actions can be treated independently in decision-making. But this assumption often does not hold. As the picture hanging example illustrates, the optimality of choosing the nail is dependent on the availability of a hammer. Therefore, our goal is to address this problem of learning the interdependence of actions.

Addressing this problem is vital for many varying action space applications. These include recommender systems where articles to recommend vary everyday and physical reasoning where decision-making must be robust to any given set of tools, objects, or skills. In this paper, we benchmark three such scenarios where learning action interdependence is crucial for solving the task optimally. These are (i) shortcut-actions in grid navigation, which can shorten the path to the goal when available, (ii) co-dependent actions in tool reasoning, where tools need other tools to be useful (like nail-hammer), and (iii) list-actions or slate-actions (Sunehag et al., 2015) in simulated and real-data recommender systems, where user response is a collective effect of the recommended list.

There are three key challenges in learning action interdependence for RL with varying action space. First, since all the actions are not known in advance, the policy framework must be flexible to work with action representations. Second, the given action space is an additional variable component, just like state observations in RL. Therefore, the policy framework must incorporate a variably sized set of action representations as part of the input. Finally, an agent's decision for each action's utility (Q-value or probability) should explicitly model its relationship with other available actions.

---

[*]Equal contribution. Corresponding author: ayushj@usc.edu
[†]Work done while at USC   [‡] AI Advisor at NAVER AI Lab
[1]Results: https://sites.google.com/view/varyingaction   Code: https://github.com/clvrai/agile

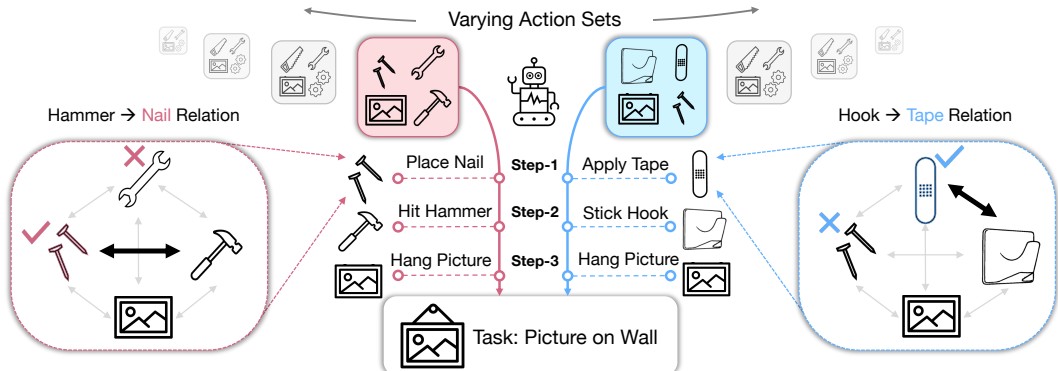

Figure 1: Picture hanging task with varying sets of tool-actions. The strategy with each action set depends on all the pairwise action relations. (Left) The agent infers that a nail and a hammer are strongly related (bold line). Thus, it takes nail-action in Step-1 because hammer-action is available for Step-2. (Right) With a different action set, the nail-action is no longer useful due to the absence of a hammer. So, the agent must use an adhesive-tape in Step-1 since its related action of the hook is available for later use. We show that a policy with GAT over actions can learn such action relations.

We propose a novel policy architecture to address these challenges: AGILE, Action Graph for Interdependence Learning. AGILE builds on the utility network proposed in Jain et al. (2020) to incorporate action representations. Its key component is a graph attention network (GAT) (Veličković et al., 2017) over a fully connected graph of actions. This serves two objectives: summarizing the action set input and computing the action utility with relational information of other actions.

Our primary contribution is introducing the problem of learning action interdependence for RL with varying action space. We demonstrate our proposed policy architecture, AGILE, learns meaningful action relations. This enables optimal decision-making in varying action space tasks such as simulated and real-world recommender systems and reasoning with skills and tools.

## 2 RELATED WORK

### 2.1 STOCHASTIC ACTION SETS

In recent work, Boutilier et al. (2018) provide the theoretical foundations of MDPs with Stochastic Action Sets (SAS-MDPs), where actions are sampled from a known base set of actions. They propose a solution with Q-learning which Chandak et al. (2020a) extend to policy gradients. An instance of SAS-MDPs is when certain actions become invalid, like in games, they are masked out from the output action probability distribution (Huang & Ontañón, 2020; Ye et al., 2020; Kanervisto et al., 2020). However, the assumption of knowing the finite base action set limits the practical applicability of SAS-MDPs. E.g., recommender agents often receive unseen items to recommend and a robotic agent does not know beforehand what tools it might encounter in the future. We work with action representations to alleviate this limitation. Furthermore, in SAS-MDPs the action set can vary at any timestep of an episode. Thus, the learned policy will only be optimal on average over all the possible action sets (Qin et al., 2020). Whereas in our setting, the action set only changes at the beginning of a task instance and stays constant over the episode. This is a more practical setup and raises the challenging problem of solving a task optimally with any given action space.

### 2.2 ACTION REPRESENTATIONS

In discrete action RL, action representations have enabled learning in large action spaces (Dulac-Arnold et al., 2015; Chandak et al., 2019), transfer learning to a different action space (Chen et al., 2019b), and efficient exploration by exploiting the shared structure among actions (He et al., 2015; Tennenholtz & Mannor, 2019; Kim et al., 2019). Recently, Chandak et al. (2020b) use them to accelerate adaptation when new actions are added to an existing action set. In contrast, our setting requires learning in a constantly varying action space where actions can be added, removed, or completely replaced in an episode. Closely related to our work, Jain et al. (2020) assume a similar setting of varying action space while training to generalize to unseen actions. Following their motivation, we use action representations to avoid assuming knowledge of the base action set. However, their policy treats each action independently, which we demonstrate leads to suboptimal performance.

### 2.3 LIST-WISE ACTION SPACE

The action space is combinatorial in list size in listwise RL (or slate RL). Commonly it has applications in recommendation systems (Sunehag et al., 2015; Zhao et al., 2017; 2018; Ie et al., 2019b; Gong et al., 2019; Liu et al., 2021; Jiang et al., 2018; Song et al., 2020). In recent work, Chen et al. (2019a) proposed Cascaded DQNs (CDQN) framework, which learns a Q-network for every index in the list and trains them all with a shared reward. For our experiments on listwise RL, we utilize CDQN as the algorithm and show its application with AGILE as the policy architecture.

### 2.4 RELATIONAL REINFORCEMENT LEARNING

Graph neural networks (Battaglia et al., 2018) have been explored in RL tasks with a rich relational structure, such as morphological control (Wang et al., 2018; Sanchez-Gonzalez et al., 2018; Pathak et al., 2019), multi-agent RL (Tacchetti et al., 2019), physical construction (Hamrick et al., 2018), and structured perception in games like StarCraft (Zambaldi et al., 2018). In this paper, we propose that the set of actions possess a relational structure that enables the actions to interact and solve tasks in the environment. Therefore, we leverage a graph attention network (Veličković et al., 2017) to learn these action relations and show that it can model meaningful action interactions.

## 3 PROBLEM FORMULATION

A hallmark of intelligence is the ability to be robust in an ever-changing environment. To this end, we consider the setting of RL with a varying action space, where an agent receives a different action set in every task instance. Our key problem is to learn the interdependence of actions so the agent can act optimally with any given action set. Figure 1 illustrates that for a robot with the task of hanging a picture on the wall, starting with a nail is optimal only if it can access a hammer subsequently.

### 3.1 REINFORCEMENT LEARNING WITH VARYING ACTION SPACE

We consider episodic Markov Decision Processes (MDPs) with discrete action spaces, supplemented with action representations. The MDP is defined by a tuple $\{\mathcal{S}, \mathbb{A}, \mathcal{T}, \mathcal{R}, \gamma\}$ of states, actions, transition probability, reward function, and a discount factor, respectively. The base set of actions $\mathbb{A}$ can be countably infinite. To support infinite base actions, we use $D$-dimensional action representations $c_a \in \mathbb{R}^D$ to denote an action $a \in \mathbb{A}$. These can be image or text features of a recommendable item, behavior characteristics of a tool, or simply one-hot vectors for a known and finite action set.

In each instance of the MDP, a subset of actions $\mathcal{A} \subset \mathbb{A}$ is given to the agent, with associated representations $\mathcal{C}$. Thereafter, at each time step $t$ in the episode, the agent receives a state observation $s_t \in \mathcal{S}$ from the environment and acts with $a_t \in \mathcal{A}$. This results in a state transition to $s_{t+1}$ and a reward $r_t$. The objective of the agent is to learn a policy $\pi(a|s, \mathcal{A})$ that maximizes the expected discounted reward over evaluation episodes with potentially unseen actions, $\mathbb{E}_{\mathcal{A} \subset \mathbb{A}} \left[ \sum_t \gamma^{t-1} r_t \right]$.

### 3.2 CHALLENGES OF VARYING ACTION SPACE

1. **Using Action Representations**: The policy framework should be flexible to take a set of action representations $\mathcal{C}$ as input and output corresponding Q-values or probability distribution for RL.

2. **Action Set as part of State:** When the action set varies, the original state space $\mathcal{S}$ is not anymore Markovian. For example, the state of a robot hanging the picture is under-specified without knowing if its toolbox contains a hammer or not. The MDP can be preserved by reformulating the state space as $S' = \{s \circ \mathcal{C}_\mathcal{A} : s \in \mathcal{S}, \mathcal{A} \subset \mathbb{A}\}$ to include the representations $\mathcal{C}_\mathcal{A}$ associated with the available actions $\mathcal{A}$ (Boutilier et al., 2018). Thus, the policy framework must support the input of $s \circ \mathcal{C}_\mathcal{A}$, where $\mathcal{A}$ is a variably sized action set.

3. **Interdependence of Actions:** The optimal choice of an action $a_t \in \mathcal{A}$ is dependent on the action choices that would be available in the future steps of the episode, $a_{t'} \in \mathcal{A}$. Recalling Figure 1, a nail should only be picked initially from the toolbox if a hammer is accessible later. Thus, an optimal agent must explicitly model the relationships between the characteristics of the current action $c_{a_t}$ and the possible future actions $c_{a_i} \forall a_i \in \mathcal{A}$.

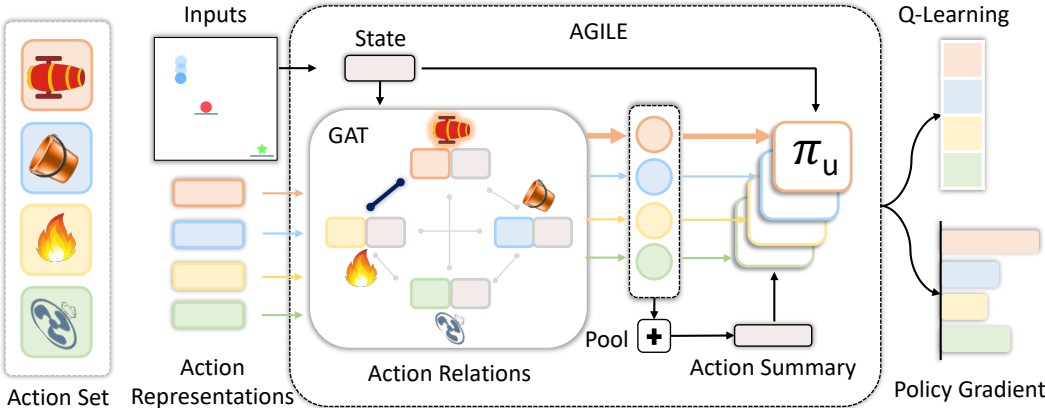

Figure 2: Given an action set, AGILE builds a complete graph where each node is composed of an action representation and the state encoding. A graph attention network (GAT) learns action relations by attending to other relevant actions for the current state. For example, the attention weight between the cannon and the fire is high because fire can activate the cannon. The GAT outputs more informed relational action representations than the original inputs. Finally, a utility network computes each action's value or selection probability in parallel using its relational action representation, the state, and a mean-pooled summary vector of all available actions' relational features.

## 4  APPROACH

Our goal is to design a policy framework that is optimal for any given action set by addressing the challenges in Sec. 3.2. We build on the utility network proposed by Jain et al. (2020) that acts in parallel on each action's representation. Our central insight is to use graph neural networks for summarizing the set of action representations as a state component and learning action relations.

### 4.1  AGILE: ACTION GRAPH FOR INTERDEPENDENCE LEARNING

We propose AGILE, a relational framework for learning action interdependence in RL with a varying action space. Given a list of action representations $\mathcal{C}$, AGILE builds a fully-connected action graph. A graph attention network (GAT) (Veličković et al., 2017) processes the action graph and learns action relations. The attention weights in the GAT would be high for closely related actions such as a nail and a hammer in Figure 1. A utility network uses the GAT's resulting relational action representations, the state, and a pooled together action set summary to compute each action's Q-value or probability logit for policy gradient methods(Figure 2).

**Action Graph**: The input to our policy framework consists of the state $s$ and a list $\mathcal{C} = [c_{a_0}, ..., c_{a_k}]$ of action representations for each action $a_i \in \mathcal{A}$. We build a fully connected action graph $\mathcal{G}$ with vertices corresponding to each available action. If certain action relations are predefined via domain knowledge, we can reduce some edges to ease training (Appendix B.1). We note that the action relations can vary depending on the environment state. For instance, a screwdriver is related to a screw for furniture repair, but a drill machine is more related when the screw is for a wall. Therefore, we join the state and action representations, $c'_{a_i} = (s, c_{a_i})$ to obtain the nodes of the graph. Sec. 6.3.2 validates that learning state-dependent action relations leads to more optimal solutions.

**Graph Attention Network**: The action graph $\mathcal{G}$ is input to a GAT. Since the graph is fully-connected, we choose an attention-based graph network that can learn to focus on the most relevant actions in the available action set. A similar insight was employed by Zambaldi et al. (2018) where the entities inferred from the visual observation are assumed to form a fully connected graph. To enable propagation of sufficient relational information between actions, we use two graph attention layers with an ELU activation in between (Clevert et al., 2015). We found a residual connection after the second GAT layer was crucial in experiments, while multi-headed attention did not help.

**Action Set Summary**: The output of the GAT is a list of relational action representations $\mathcal{C}^R = \{c_{a_0}^R, ...c_{a_k}^R\}$ which contain the information about the presence of other available actions and their relations. To represent the input action set as part of the state, we compute a compact action set summary by mean-pooling the relational action features, $\bar{c}^R = \frac{1}{K} \sum_{i=1}^{K} c_{a_i}^R$.

**Action Utility**: To use the relational action representations with RL, we follow the utility network architecture $\pi_u$ from Jain et al. (2020). It takes the relational action representation, the state, and the action set summary as input for each available action in parallel. It outputs a utility score $\pi_u(c_a^R, s, \bar{c}^R)$ for how useful an action $a$ is for the current state and in relation to the other available actions. The utility scores can be used as a Q-value directly for value-based RL or as a logit fed into a softmax function to form a probability distribution over the available actions for policy-based RL.

## 4.2 Training AGILE framework with Reinforcement Learning

The AGILE architecture can be trained with both policy gradient and value-based RL methods.

**Policy Gradient (PPO)** : For every action decision step, we take the output action utility and use a softmax over all available actions to get a probability distribution. We use PPO (Schulman et al., 2017) to train the policy. PPO requires a value function $V(s')$ as a baseline. We represent the effective state as the concatenation of the state encoding $s$ and the action set summary $\bar{c}^R$ inferred from the GAT, $s' = (s, \bar{c}^R)$. An important implementation detail to make the training faster and stable was to *not* share the GAT weights used for the actor $\pi(a|s', c_a^R)$ and the critic $V(s')$ networks.

**Value-based RL (DQN)** The output action utility can be directly treated as the $Q(s', a)$ value of the action $a$ at the current effective state $s'$ with the available action set $\mathcal{A}$. This can be trained using standard Deep Q-learning Bellman backup (Mnih et al., 2015).

**Listwise RL (CDQN)**: For tasks with listwise actions, we follow the Cascaded DQN (CDQN) framework of Chen et al. (2019a). The main challenge is that building the action list all at once is not feasible due to a combinatorial number of possible list-actions. Therefore, the key is to build the list incrementally, one action at a time. Thus, each list index can be treated as an individual action decision trained with independent Q-networks. We replace the Q-network of CDQN with AGILE to support a varying action space. Sharing the weights of the cascaded Q-networks led to better performance. Algorithm 1 provides complete details on CDQN for listwise AGILE.

## 5 Environments

We evaluate AGILE on three varying action set scenarios requiring learning action interdependence: (i) shortcut actions in goal-reaching, which can shorten the optimal path to the goal in a 2D Grid World when available, (ii) co-dependent actions in tool reasoning, which require other tools to activate their functionality, and (iii) list-actions in simulated and real-data recommender systems where the cumulative list affects the user response. Figure 3 provides an overview of the tasks, the base and varying action space, and an illustration of the action interdependence. More environment details such as tasks, action representations, and data collection are present in Appendix A.

### 5.1 Dig Lava Grid Navigation

We modify the grid world environment (Chevalier-Boisvert et al., 2018) where an agent navigates a 2D maze with two lava rivers to reach a goal. The agent always has access to four direction movements and a *turn-left* skill (Figure 5). There are two additional actions randomly sampled out of four special skills: *turn-right, step-forward, dig-orange-lava* and *dig-pink-lava*. If the agent enters the lava, it will die unless it uses the matching *dig-lava* skill to remove the lava in the immediately next timestep. Thus, when available, *dig-lava* skills can be used to create shortcut paths to the goal and receive a higher reward. We use PPO for all experiments in this environment.

### 5.2 Chain REAction Tool Environment: CREATE

The CREATE environment (Jain et al., 2020) is a challenging physical reasoning benchmark with a large variety of tools as actions and also supports evaluation with unseen actions. The objective is to sequentially place tools to help push the red ball towards the green goal. An action is a hybrid of a discrete tool-selection from varying toolsets and a continuous $(x, y)$ coordinate of tool-placement on the screen. An auxiliary policy network decides the tool-placement based on the effective state $s'$ as input, following Jain et al. (2020). To emphasize action relations, we augment the environment with special activator tools (e.g., fire) that general tools (e.g., cannon) need in contact for being functional. Thus, a general tool can be useful only if its activator is also available. Action representations of general tools encode their physical behavior (Jain et al., 2020), while those of activator tools are one-hot vectors. We train AGILE and the auxiliary policy jointly with PPO.

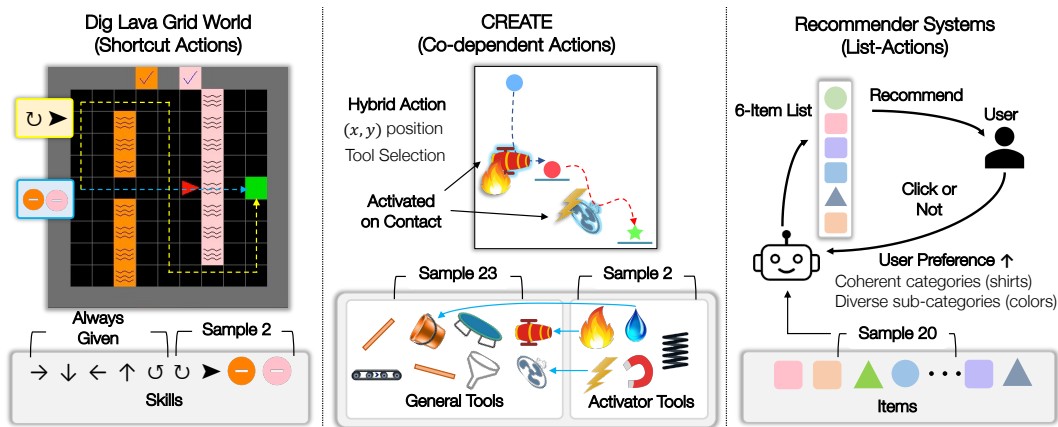

Figure 3: **Environment Setup.** (Left) In Grid World, the red agent must avoid orange and pink lava to reach the green goal. The *dig-lava* skills enable shortcut paths to the goal when available. (Middle) In CREATE, tools are selected and placed sequentially to push the red ball to the green goal. General tools (cannon, fan) require activator tools (fire, electric) to start functioning. The choice of general tools depends on what activators are available and vice-versa. (Right) In simulated and real-data recommender systems, the agent selects a list of items. Lists with coherent categories and complementary sub-categories improve user satisfaction (e.g., click likelihood).

## 5.3 RECOMMENDER SYSTEMS

Recommender system (RecSys) is a natural application of varying action space RL — for instance, news articles or videos to recommend are updated daily. Action interdependence is distinctly apparent in the case of listwise actions. A recommended list of diverse videos is more likely to get a user click than videos about the same thing (Zhou et al., 2010). In this work, we experiment with the listwise metric of Complementary Product Recommendation (CPR) (Hao et al., 2020).

**Complementary Product Recommendation (CPR)** is a scenario where user response for a recommended list is more favorable if the list is diverse at a low level but coherent at a high level. In our experiments, each item has a primary category (such as shirt or pant) and a subcategory (such as its color). We define the CPR of an item-list as, $\frac{\text{Entropy of subcategory}}{\text{Entropy of category}}$. This encourages diversity in subcategories (colors) and similarity in the main category (all shirts). We maximize CPR (i) implicit in the click-behaviors of simulated users and (ii) explicit in the reward computed on real-world data.

### 5.3.1 SIMULATED RECOMMENDER SYSTEM: RECSIM

We use RecSim (Ie et al., 2019a) to simulate user interactions and extend it to the listwise recommendation task. We have a base action set of 250 train and 250 test items, and 20 items are sampled as actions for the agent in each episode. The agent recommends a list-action of size six at each step. We assume a fully observable environment with the state as the user preference vector and the action representations as item characteristics. The objective implicitly incorporates CPR by boosting the probability of a user clicking any item proportional to the list CPR. The implicit CPR objective exemplifies realistic scenarios where the entire list influences user response. One way to optimize CPR is to identify the most common category in the available action set and recommend most items from that category. Such counting of categories requires relational reasoning over all items available in the action set. We train CDQN-based models to maximize the number of clicks in a user session.

### 5.3.2 REAL-DATA RECOMMENDER SYSTEM

We collect four-week interaction data in a listwise online campaign recommender system. Users are represented by attributes such as age, occupation, and localities. Item attributes include text features, image features, and reward points of campaigns. We train a VAE (Kingma & Welling, 2013) to learn item representations. We create a representative RL environment by training two click-estimation models using data from the first two weeks for training and the last two weeks for evaluation. The training environment consists of 68,775 users and 57 items, while testing has 82,445 users and 58 items, with an overlap of 30 items. The reward function combines the user-click and CPR value of the list. The explicit CPR reward is a representative scenario for when the designer has listwise objectives in addition to user satisfaction. We train with CDQN and report the test reward.

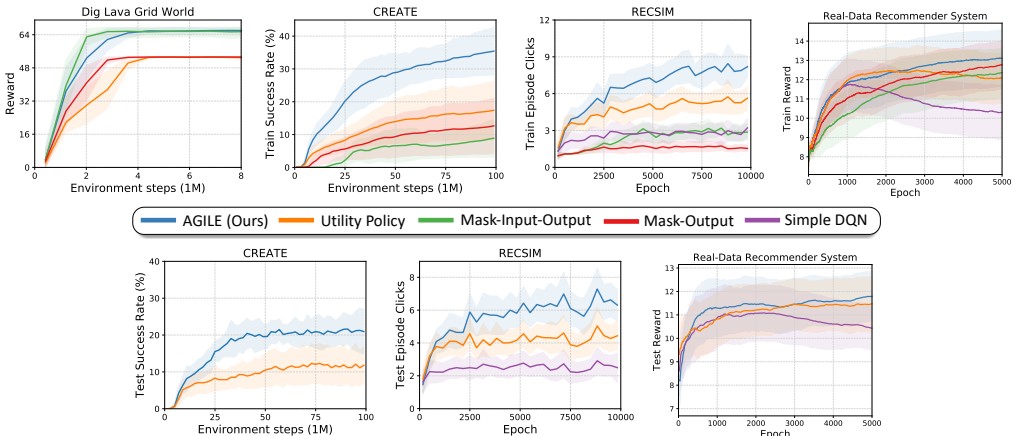

Figure 4: We evaluate AGILE against baselines on training actions (top) and unseen testing actions (bottom). Generalization is enabled by continuous action representations, except Grid World (Appendix A). All architectures share the same RL algorithm (PPO or CDQN). The results are averaged over 5 random seeds, and the seed variance is shown with shading. AGILE outperforms all baselines that assume a fixed action set (cannot generalize) or treat actions independently (suboptimal).

## 6 EXPERIMENTS

We design experiments to answer the following questions in the context of RL with varying action space: (1) How effective is AGILE compared to prior works that treat actions independently or assume a fixed action set? (2) How effective are AGILE's relational action representations for computing action set summary and action utility score? (3) Does the attention in AGILE represent meaningful action relations? (4) Is attention necessary in the graph network of AGILE? (5) Is learning state-dependent action relations important for solving general varying action space tasks?

### 6.1 EFFECTIVENESS OF AGILE IN VARYING ACTION SPACES

We evaluate AGILE against baselines from prior work in varying action space, which either assume a fixed action set or act independently of the other actions. We ablate the importance of relational action features by replacing them with the original action representations and compute the action set summary in different ways. The Appendix details the comparison (C.1) and visualization (Figure 17) of all baselines and ablations, hyperparameter-tuning (D.2,D.3) and network designs(C.3).

#### 6.1.1 BASELINES

- **Mask-Output** (No action representations, No input action set): Assumes a fixed action space output. Q-values or policy probabilities are masked out for unavailable actions. It represents prior work on SAS-MDP: Boutilier et al. (2018); Chandak et al. (2020a); Huang & Ontañón (2020).

- **Mask-Input-Output** (No action representations): Augments *Mask-Output* with an extra input of action set via a binary availability vector: having 1s at available action indices and 0s otherwise.

- **Utility-Policy** (No input action set): Jain et al. (2020) propose a parallel architecture to compute each action's utility using action representations. But, it ignores any action interdependence.

- **Simple DQN** (No cascade, No input action set): A non-cascaded DQN baseline for listwise RL that selects top-K items instead of reasoning about the entire list. Thus, it ignores two action interdependences: (i) on other items in the list and (ii) on other available actions.

#### 6.1.2 ABLATIONS

- **Summary-LSTM**: A Bi-LSTM (Huang et al., 2015) encodes the list of action representations.

- **Summary-Deep Set**: A deep set architecture (Zaheer et al., 2017) with mean pooling is used to aggregate the available action representation list.

- **Summary-GAT**: The relational action representations output from the GAT in AGILE is used only for the action set summary but not for the utility network. This does not scale to tasks with many diverse inter-action relations because the summary vector has a limited modeling capacity.

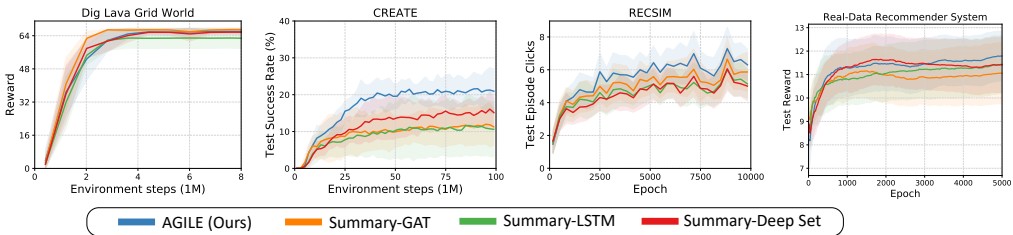

Figure 5: Test performance of AGILE against ablations with utility network using only action set summary, but not the relational action representations. The difference is most pronounced in CRE-ATE, where there are several diverse action (tool) relations for each action decision.

### 6.1.3   RESULTS

**Baselines**: Figure 4 shows baseline results on train and test actions. Grid world has no unseen actions, so we report only train reward. **Grid World**: all methods learn to reach the goal with perfect success but achieve different rewards due to the length of the path. The methods which do not take action set as input, *Mask-Output* and *Utility Policy*, resort to the safe strategy of going around the lava rivers (Figure 6(b)). This is because the agent must enter the lava and then dig it to take a short-cut. So before taking the *move-right* action into lava, the agent must know whether that *dig-lava* skill is available. **CREATE**: *Mask* baselines do not support generalization due to fixed action set assumption and train poorly as they do not exploit the structure in action representations. *Utility policy* outperforms the other baselines since it can exploit learning with action representations. Finally, AGILE outperforms all the baselines, demonstrating that relational knowledge of other available actions is crucial for an optimal policy. **RecSim** and **Real RecSys**: result trends are consistent with CREATE, but less pronounced for Real RecSys. Additionally, DQN is worse than CDQN-based architectures because the top-K greedy list-action ignores intra-list dependence.

**Ablations**: Figure 5 shows ablation results on test actions. **Grid World**: all ablations utilize the action set summary as input, aggregated via different mechanisms. Thus, they can identify which *dig-lava* skills are available and enter lava accordingly to create shortcuts. In such small action spaces with simple action relations, summary-ablations are on par with AGILE. This trend also holds for **RecSim** and **Real RecSys**, where the summary can find the most common category and its items are then selected to maximize CPR (e.g., Figure 6(c)). Therefore, we observe only $5-20\%$ gains of AGILE over the ablations. To test the consistency of results, we further evaluate two more RecSim tasks. (i) Direct CPR: the agent receives additional explicit CPR metric reward on top of click/no-click reward (Sec. B.3), and (ii) pairing environment: the task is to recommend pairs of associated items based on predefined pairings (Sec. B.4). We reproduce the trend that AGILE $>=$ ablations. However, the difference is most pronounced $(30-50\%)$ in **CREATE**, where each action decision relies on relations between various tools and activators. In Figure 6(a), the decision of *Spring* relates with all other tools that spring can activate, and the decision of *trampoline* relates with its activator, *Spring*. In ablations, the action set summary must model all the complex and diverse action relations with a limited representation capacity. In contrast, the relational action representations in AGILE model each action's relevant relations, and the summary models the global relations.

### 6.2   DOES THE ATTENTION IN AGILE LEARN MEANINGFUL ACTION RELATIONS?

In Figure 6, we analyze the agent performance qualitatively. (a) In CREATE, at $t = 0$, the selected action *spring* in AGILE's GAT attends to various other tools, especially the tools that get activated with *spring*, such as *trampoline*. At $t = 1$, the *trampoline* tool is selected with strong attention on *spring*. This shows that for selecting the *trampoline*, the agent checks for its activator, *spring*, to ensure that it is possible to place *spring* before or after the trampoline. (b) In Grid World, we visualize the inter-action attention in *Summary-GAT*'s summarizer. We consider the case where both $dig-lava$ skills are available. The agent goes right, digs the orange lava, and is about to enter the pink lava. At this point, the *Right* action attends with a large weight to the $Dig - Pink$ skill, checking for its presence before making an irreversible decision of entering the lava. In contrast, the *Utility Policy* always follows the safe suboptimal path as it is blind to the knowledge of dig-skills before entering lava. (c) In RecSim, we observe that the agent can maximize the CPR score by selecting 5 out of 6 items in the list from the same primary category. In contrast, *Utility Policy* cannot determine the most common available category and is unable to maximize CPR.

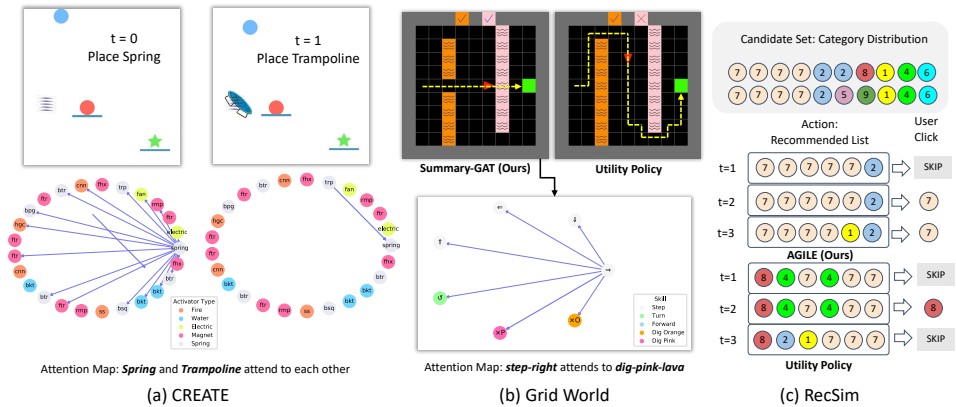

Figure 6: Qualitative Analysis. (a,b) The attention maps from GAT show the reasoning behind an action decision. The nodes show available actions, and edge widths are proportional to attention weight (thresholded for clarity). (b) Utility Policy learns the same suboptimal solution for any given action set, while Summary-GAT (like AGILE) adapts to the best strategy by exploiting dig-skills. (c) AGILE can optimize CPR by identifying the most common item category available, unlike Utility Policy. We provide qualitative video results for Grid World and CREATE on the project page https://sites.google.com/view/varyingaction.

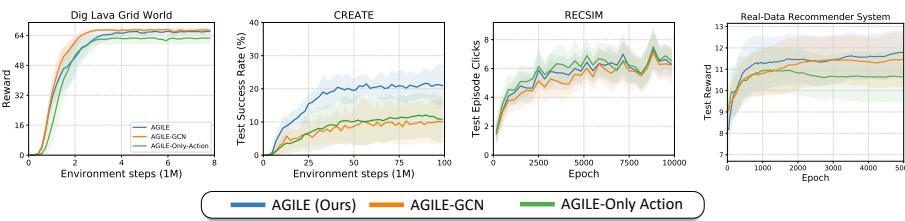

Figure 7: Additional Analyses. (i) GAT v/s GCN (ii) state-action relations v/s action-only relations.

## 6.3 ADDITIONAL ANALYSES

### 6.3.1 IMPORTANCE OF ATTENTION IN THE GRAPH NETWORK

We validate the choice of using graph attention network as the relational architecture. In Figure 7, we compare GAT against a graph convolutional network (GCN) (Kipf & Welling, 2016) to act over AGILE's action graph. We observe that GCN achieves optimal performance for the grid world and RecSys tasks. GCN can learn simple action relations even though the edge weights are not learned. However, it suffers in CREATE and RecSim-pairing (Figure 16), where the action relations are diverse and plenty. Moreover, we believe that the attention in GAT makes the graph sparse to ease RL training, which in contrast, is difficult in a fully-connected GCN.

### 6.3.2 IMPORTANCE OF STATE-DEPENDENT LEARNING OF ACTION RELATIONS

We evaluate a version of AGILE where the GAT only receives action representations as input and no state. Thus, the action relations are inferred independently of the state. Figure 7 shows a drop in performance for Grid World and CREATE, where the relevant action relations change based on the state. However, this effect is less apparent on RecSim because CPR requires only knowing the most common category, independent of user state.

## 7 CONCLUSION

We present AGILE, a policy architecture for leveraging action relations for reinforcement learning with varying action spaces. AGILE builds a complete graph of available actions' representations and utilizes a graph attention network to learn the interdependence between actions. We demonstrate that using the knowledge of available actions is crucial for optimal decision-making and relational action features ease learning in four environments, including a real-data recommendation task.

ACKNOWLEDGEMENTS

This project was supported by NAVER AI Lab and USC. The authors are grateful to Youngwoon Lee for help with ideation and guidance. We thank Xingdong Zuo for help with data extraction in real-data RecSys. We appreciate the fruitful discussions with Shao-Hua Sun, Karl Pertsch, Jesse Zhang, Grace Zhang, and Jisoo Lee.

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

# A   Environment Details

## A.1   Dig Lava Grid Navigation

The grid world environment, introduced in Sec. 5.1, requires an agent to reach a goal by navigating a 2D maze with two lava rivers using a variable set of skills.

**State**: The state space is a concatenation of two flattened 9x9 grids:

1. Environment: each grid element is an integer ID representing whether the cell is empty, orange-lava, pink-lava, goal, or sub-goal
2. Agent: an empty grid except for 1 at agent-coordinates.

**Termination**: An episode is terminated in success when the agent reaches the goal and in failure when it stays in lava for two consecutive timesteps or after a total of 50 timesteps. Note that the agent can leave a lava cell only using the dig lava skill of the corresponding color.

**Actions**: The base action set is a fixed set of 9 skills and in each episode, a set of 7 skills is given to the agent. 5 skills are always available: *move-right*, *move-down*, *move-left*, *move-up*, and *turn-left*. The other 2 skills are randomly sampled from a set of 4 skills: *turn-right*, *step-forward*, *dig-orange-lava*, and *dig-pink-lava*.

**Reward**: The agent receives a large goal reward on reaching the goal. There are two subgoal rewards for a successful crossing of each lava column (i.e. column 4 and column 8) for the first time. The goal and subgoal rewards are discounted based on the number of action steps taken to reach that location, thus rewarding shorter paths. To further encourage shorter paths, successful lava digging is rewarded. A small exploration reward is added whenever the agent visits a new cell in the episode. The exploration reward is accumulated and subtracted when the agent reaches a subgoal or a goal to ensure that the exploration reward does not hinder learning short paths. Thus,

$$
\begin{aligned}
R(s, a) &= \mathbb{1}_{Goal} \cdot \left[ R_{\text{Goal}} \left( 1 - \lambda_{\text{Goal}} \frac{N_{\text{current steps}}}{N_{\text{max steps}}} \right) - R_{\text{Exploration}} \, N_{\text{steps from prev subgoal}} \right] + \\
&\quad \mathbb{1}_{Subgoal} \cdot \left[ R_{\text{Subgoal}} \left( 1 - \lambda_{\text{Subgoal}} \frac{N_{\text{current steps}}}{N_{\text{max steps}}} \right) - R_{\text{Exploration}} \, N_{\text{steps from prev subgoal}} \right] + \\
&\quad \mathbb{1}_{\text{Successful Dig}} \cdot R_{\text{Dig}} \; + \; \mathbb{1}_{\text{New State}} \cdot R_{\text{Exploration}}
\end{aligned}
\tag{1}
$$

where $R_{\text{Goal}} = 100, \; R_{\text{Subgoal}} = 0.5, \; R_{\text{Exploration}} = 0.01, \; R_{\text{Dig}} = 0.01,$ $\lambda_{\text{Goal}} = 0.99, \; \lambda_{\text{Subgoal}} = 0.9, \; N_{\text{max steps}} = 50$

**Action Representations**: The action representations are 11-dimensional vectors manually defined using a mix of one-hot vectors, as shown in Table 1. Dimensions 1-5 identify the category of skills (movement, elemental, dig-orange, dig-pink), 6-7 distinguish movement skills (right, down, left, up), 8-9 are always 0 (originally meant for diagonal skills), 10-11 are used to distinguish elemental skills (turn-left, turn-right, move-forward).

| Category | Skill | Action Representation | | | | | | | | | | |
|---|---|---|---|---|---|---|---|---|---|---|---|---|
| Movement | move-right | 0 | 0 | 0 | 0 | 1 | -1 | -1 | 0 | 0 | 0 | 0 |
| | move-down | 0 | 0 | 0 | 0 | 1 | -1 | 1 | 0 | 0 | 0 | 0 |
| | move-left | 0 | 0 | 0 | 0 | 1 | 1 | -1 | 0 | 0 | 0 | 0 |
| | move-up | 0 | 0 | 0 | 0 | 1 | 1 | 1 | 0 | 0 | 0 | 0 |
| Elemental | turn-left | 0 | 0 | 1 | 0 | 0 | 0 | 0 | 0 | 0 | -1 | 1 |
| | turn-right | 0 | 0 | 1 | 0 | 0 | 0 | 0 | 0 | 0 | 1 | -1 |
| | move-forward | 0 | 0 | 1 | 0 | 0 | 0 | 0 | 0 | 0 | 1 | 1 |
| Digging | dig-orange | 0 | 1 | 0 | 0 | 0 | 0 | 0 | 0 | 0 | 0 | 0 |
| | dig-pink | 1 | 0 | 0 | 0 | 0 | 0 | 0 | 0 | 0 | 0 | 0 |

Table 1: Action representations for the skills used in Dig Lava Grid Navigation environment.

### A.2 CHAIN REAction TOOL ENVIRONMENT (CREATE)

The CREATE environment (Sec. 5.2) requires an agent to place tools in a physics-based puzzle to make the target ball reach the goal position. We follow all the base settings of the CREATE Push Environment from Jain et al. (2020). We add the functionality of new activator tools and reduce the number of available actions per episode from 50 to 25.

**State**: The agent receives the past three gray-scale frames, which are stacked channel-wise to make a 84x84x3 input.

**Termination**: The episode ends in success when the agent accomplishes the goal and in failure when there is no remaining movement in the game or after 30 timesteps.

**Actions**: Each original tool from CREATE is now associated with newly added *activator tools* as shown in Figure 8. Thus, we add 5 activator tools to the 2110 general tools, which are variations in angle, size, friction, or elasticity of the tools shown in Figure 8. Unlike Jain et al. (2020), we remove the No-Operation action. Activator tools are pass-through tools and only serve the function of activating their corresponding general tools when placed in contact.

We split the general tool space into 1098 tools for training, 507 tools for validation, and 507 tools for testing. All 5 activator tools are available for sampling during training, validation, and testing. In each episode, 23 general tools and 2 activator tools are randomly sampled and made available to the agent. The agent outputs in a hybrid action space consisting of (1) the discrete tool selection and (2) $(x, y)$ coordinates of tool position.

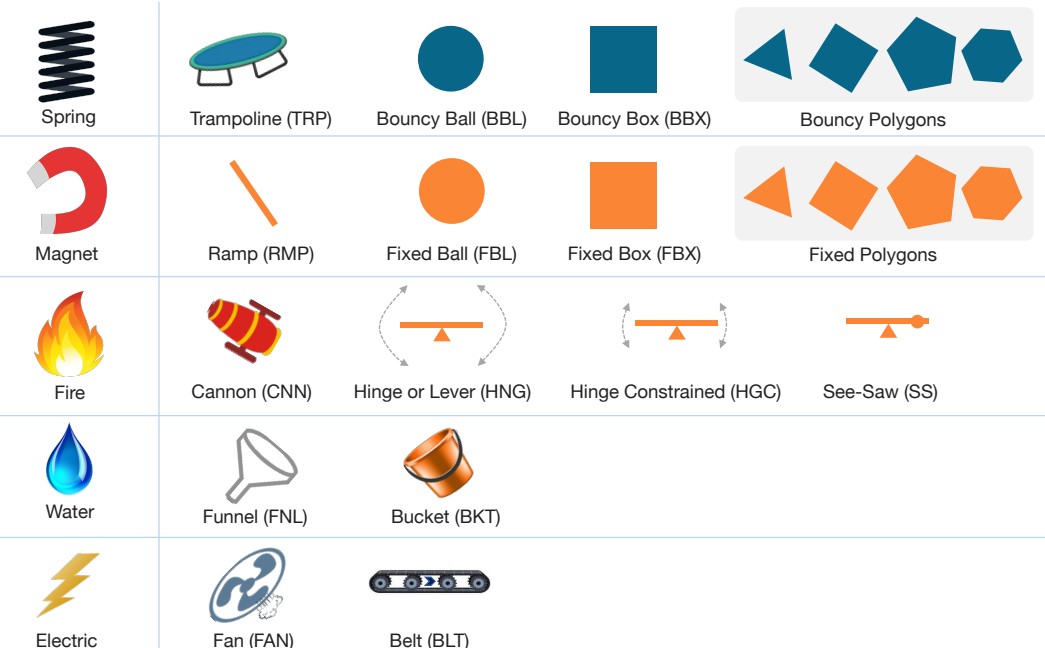

Figure 8: **CREATE Activator Mapping**: Original CREATE tools (right) are activated by the respective newly introduced activator tools (left). E.g., a *Cannon* tool (abbreviated as CNN in attention maps) placed on the environment will only be functional when a *Fire* tool is placed in contact with it. Other objects are not affected by non-functional tools - they simply pass through.

**Reward**: We adopt the reward structure from Jain et al. (2020). CREATE Push is a sparse reward environment with rewards for reaching the goal and making the target ball move. There is an additional *alive* reward to continue the episode. Placements outside the scene are penalized. The original CREATE environment penalizes overlapping tools. However, since overlapping activators and general tools is required for solving tasks, we modify the overlap penalty to apply only when either both are general tools or both are activator tools. Thus, the agent receives the following reward:

$$R(s,a) = R_{\text{alive}} + \mathbb{1}_{\text{Goal}} \cdot R_{\text{Goal}} + \mathbb{1}_{\text{target hit}} \cdot R_{\text{target hit}} + \mathbb{1}_{\text{invalid}} \cdot R_{\text{invalid}} + \mathbb{1}_{\text{overlap}} \cdot R_{\text{overlap}} \tag{2}$$

where $R_{\text{alive}} = 0.01$, $R_{\text{Goal}} = 10.0$, $R_{\text{target hit}} = 1$, and $R_{\text{invalid}} = R_{\text{overlap}} = -0.01$.

**Action Representations**: Each action representation is a 134-dimensional vector, a concatenation of two 128-D and 6-D vectors, as shown in Table 2. The 128-D vector corresponds to the learned characteristics of the tool, and the 6-D vector is a binary vector denoting whether the tool is a general tool, an activator tool, or a no-op tool (no-op is disabled for experiments).

For general tools, Jain et al. (2020) obtain action representations using a Hierarchical VAE. They encode a set of tool observations into a latent representation. Each tool is made to interact with a probing ball launched from various angles, positions, and speeds. The tool characteristics can be inferred from this collection of tool interactions. We utilize these 128-D learned tool representations from Jain et al. (2020) for the general tools and pad them with a 6-D zero-vector. Generalization to unseen tools is possible because a trained agent can utilize the 128-D tool embedding to extract the relevant characteristics of any given tool. The agent must infer which activator a general tool is associated with using its tool embedding.

For activator tools, the first 128 dimensions are always zero, and the final 6 dimensions correspond to a one-hot vector, each for {no-op, Fire, Water, Electric, Magnet, Spring}.

| Tool | Action Representation | | | | | | |
|---|---|---|---|---|---|---|---|
| General Tools | Learned 128-D tool characteristics | 0 | 0 | 0 | 0 | 0 | 0 |
| No-Op | 128-D Zero Vector | 1 | 0 | 0 | 0 | 0 | 0 |
| Fire | 128-D Zero Vector | 0 | 1 | 0 | 0 | 0 | 0 |
| Water | 128-D Zero Vector | 0 | 0 | 1 | 0 | 0 | 0 |
| Electric | 128-D Zero Vector | 0 | 0 | 0 | 1 | 0 | 0 |
| Magnet | 128-D Zero Vector | 0 | 0 | 0 | 0 | 1 | 0 |
| Spring | 128-D Zero Vector | 0 | 0 | 0 | 0 | 0 | 1 |

Table 2: Action representations for the tools in CREATE environment.

### A.3 RECSIM

The simulated RecSys environment (RecSim in Sec. 5.3.1), requires an agent to select a list of items that match the user's interest out of a variety of recommendable items. We simulate users that have a preference over high-CPR lists (Sec 5). The agent's task is to infer this preference from user clicks and recommend a list of items to optimize both user interest and CPR.

**State:** The state is represented by the user interest embedding ($e_u \in \mathbb{R}^n$ where $n$ denotes the number of categories of items) in categories that transitions over time as the user consumes different items upon click. So, when the user clicks an item with the corresponding item embedding($e_i \in \mathbb{R}^n$) then the user interest embedding($e_u$) will be updated as follows,

$$\Delta(e_u) = (-y|e_u| + y) \cdot (1 - e_u), \text{ for } y \in [0, 1]$$
$$e_i \leftarrow e_u + \Delta(e_u) \text{ with probability}[e_u^T e_i + 1]/2$$
$$e_u \leftarrow e_u - \Delta(e_u) \text{ with probability}[1 - e_u^T e_i]/2$$

This essentially pulls the user's preference towards the item that was clicked.

**Action**: The base action set is a set of 500 items (250 for each train and test action set), and in each episode, a sampled subset of size 20 is given to the agent. To simulate the varying action space environment, we implemented the **most common category sampling** method to form the candidate-set. Here, the majority of items are sampled from one common category, and the remaining items are sampled from other categories. In this way, identifying the most common category is crucial to recommend a coherent list of items. Note that if we just sampled items uniformly across all categories, then CPR maximization would be less interesting as no single category has the potential to fill the entire list of items (i.e., achieve maximum CPR).

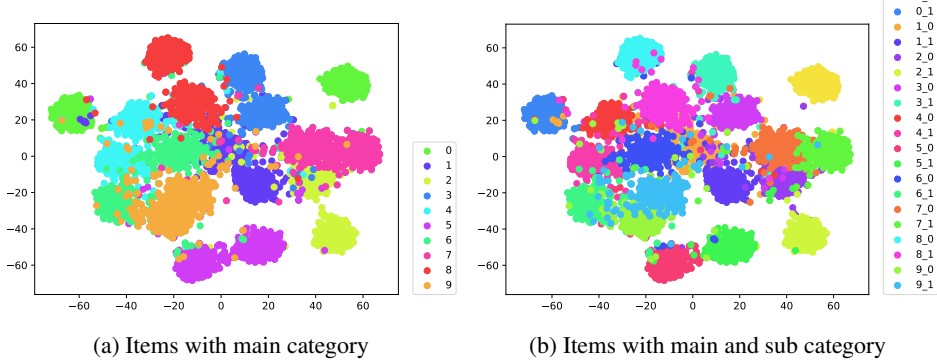

(a) Items with main category        (b) Items with main and sub category

Figure 9: t-SNE visualization of synthetically generated items in RecSim.

**Reward**: The base reward is a simulated response (e.g., clicks) from users (the user model (Ie et al., 2019a) stochastically skips or clicks an item in the list based on the user interest embedding). To better simulate the realistic scenario of user preference being affected by CPR of a presented list, we implement additional features in the user model:

$$\text{score}_{item} = \alpha_{user} * \langle e_u, e_i \rangle + \alpha_{metric} * m$$

$$p_{item} = \frac{e^{s_{item}}}{\sum e^{s_{item}}}$$

$$R = f_{\text{click\_or\_skip}}(p_{item})$$

where, $e_u, e_i \in \mathbb{R}^n$ are the user and item embedding, respectively, $\langle \cdot, \cdot \rangle$ is the dot product notation and $\alpha_{user}, \alpha_{metric}, m \in \mathbb{R}$ where $m$ denotes the list-metric (e.g., CPR-score). So, given the score $\text{score}_{item}$ of an item, the user model computes the click likelihood through a softmax function over all items and a predefined skip-item followed by a categorical distribution ($f_{\text{click\_or\_skip}}$). It takes the computed likelihood and outputs either click (reward=1) or skip (reward=0) as user feedback.

**Action Representations**: Originally, Ie et al. (2019a) use the discrete representation of items based on the one-hot encoding of item-category. However, this does not support generalization over items (actions). Therefore, we implement continuous item representations sampled from a Gaussian Mixture Model (GMM) with centers around each item category. Each item category has two sub-categories for items, which are also clustered. This ensures that the action representation contains information about the primary and sub-categories. Fig. 9 shows the t-SNE visualization of action representations for the 500 items, based on a GMMs. There are 10 main categories that each have two sub-categories. Figure 9a shows item representations labeled according to the primary category. Figure 9b shows item representations clustered according to sub-category.

### A.4 REAL-DATA RECOMMENDER SYSTEM

The real-data RecSys environment (Sec. 5.3.2) requires an agent to select a list of items that match the user's interest out of a variety of recommendable items. We experiment with the scenario in which domain engineers want an RL agent's list-actions to conform to user preference while optimizing a listwise metric. Since having a high CPR is correlated with better user response (Hao et al., 2020), we use CPR as the additional listwise metric. Thus, we reward the agent based on (i) user models trained from real data and (ii) CPR of the recommended list of items. We collected the dataset from two different periods, two weeks in late August in 2021 as the offline period (used for training) and the following two weeks from early September in 2021 as the online period (used for evaluation).

**State**: The state is a concatenation of a sequence of historical user interactions. Concretely, a set of item representations (32 dimensional real vectors) of the three most recent clicked items (i.e., $32 \times 3$ matrix) are appended individually to the user attributes (137 dimensional real vectors) such as age, occupation, and localities. Therefore, the state is in the form of $507 = 169 \times 3 = (32 + 137) \times 3$. To act optimally, the agent must extract the useful representation of the user preference through this historical observation.

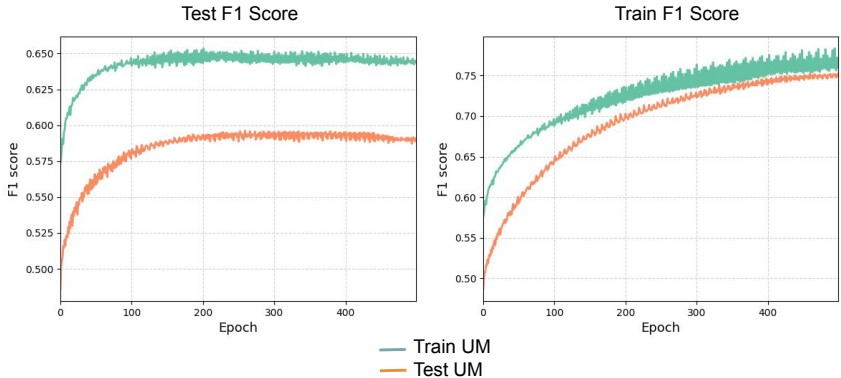

Figure 10: Real-Data Recommender System: F1 Score for Training/test user models. (Left) The online user model is trained on online-training data and evaluated on both online-training (green) and online-held-out (orange) data. (Right) The offine user model is trained on offline-training data and evaluated on both offline-training (green) and offline-held-out data (orange). Thus, the disparity between online data training and evaluation curves (left) shows that it is hard to train the online user model. In contrast, the offline user model generalizes reasonably well (right).

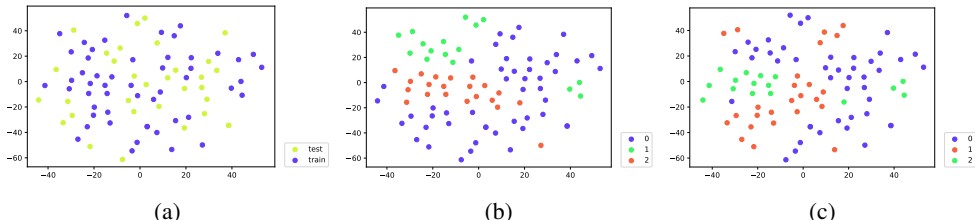

Figure 11: (a) t-SNE visualization of split of train and test item representations. This shows that the train and test items are within the same distribution (b) Visualizations of item representations clustered according to the main category. This shows that item representations contain information about the main category, which is necessary to maximize CPR. (c) Item distribution labeled according to sub category, which is another information necessary for CPR.

**Action**: The base action set is a set of 85 items, and in each episode, a sampled subset of size 20 is given to the agent. We employed the same sampling methods as in Sec. A.3. So, the agent needs to select a list of 6 items given the sampled candidate set of 20 items.

**Reward**: The base reward is a simulated response (click or skip) from the user model trained from real-world data. We add the CPR metric (between 0 to 1) to this click reward (1 or 0). The agent's objective is to optimize user preference and the CPR of the list of items it recommends.

*Training of User models*: We follow the two different periods (offline and online) in the data extraction procedure. Thus, we trained two different user models ($f_{user} : \mathcal{S} \times \mathcal{A}^{\text{list-size}} \to \mathbb{R}$) to (i) train the agents offline and (ii) evaluate them with online users. The result of training those user models can be found in Figure 10 (a). Thus, the reward is computed as follows; $R = f_{user}(e_{user}, e_{item}) + m$ where $f_{user}$ is either the offline user model or the online user model that provides us with the simulated response of users (e.g., click or skip). The user model architectures are described in Sec. C.3.5.

**Action Representation:** In the previous work CDQN (Chen et al., 2019a)), the authors found it useful to characterize the items by wide and deep features. For example, their movie recommendation task considered the text description as the wide feature and the movie category as the deep feature. And they utilized the Wide and Deep Network (Cheng et al., 2016) to get useful representations of items. Following their work, we used the wide and deep network to pretrain the item embedding given the rich item features in the real-world data. However, we empirically observed that employing VAEs for each wide and deep feature of items leads to the better-segregated representation of items. Therefore, given an item instance, we separate the raw item attributes into the deep attributes (e.g., reward points of campaigns) and the wide attributes (e.g., text description),

which are then fed into the VAE based wide and deep network to get the compressed representation by combining the wide and the deep features together. See Sec C.3.4 for more details about the network architecture. Thus, each action representation is a 32-dimensional vector encoded by a VAE based Wide and Deep network. The learned representations of items are visualized based on the train-test split, distribution of the main category, and distribution of sub-category in Figure 11.

# B FURTHER EXPERIMENTAL RESULTS

## B.1 EFFECT OF USING DOMAIN KNOWLEDGE IN ACTION GRAPH EDGES

To show how to incorporate domain knowledge about action relations into the action graph (discussed in Section 4), we use the Dig Lava Grid World environment. Specifically, we use the knowledge that directional actions are always available. Thus, the only relevant action relations are the ones with the variable actions, i.e., 2 out of 4 actions (including both dig-lava skills). So, while building the action graph, we only keep connections to and from the 2 variable actions instead of having a fully-connected structure. The rest of the AGILE architecture and algorithm stays the same. This reduces the number of bidirectional edges from $7^2 = 49$ to 29 by removing 20 edges between the always available actions. As shown in Figure 12, the learning speed of AGILE is slightly accelerated, and the seed variance has reduced. We expect domain knowledge to help efficiency even more, when the action space is large since the edges scale quadratically.

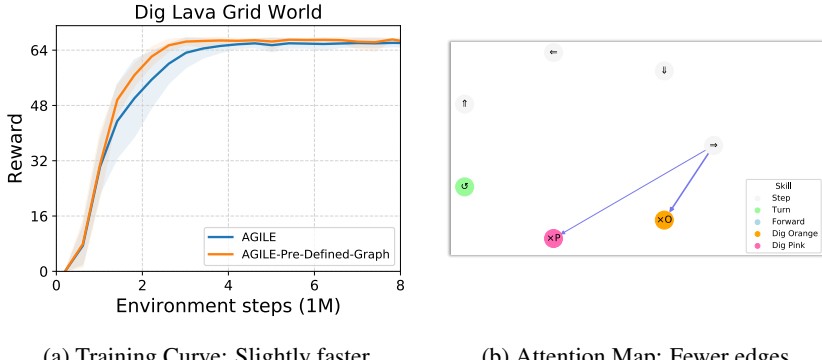

(a) Training Curve: Slightly faster      (b) Attention Map: Fewer edges

Figure 12: Effect of using domain knowledge to predefine the possible relations via edges in the action graph of AGILE. We know that in the Grid Navigation task, only the action relations with respect to variable actions are important. Thus, we remove all the other edges. This makes the learning slightly faster and more stable as shown by a reduction in seed variance. (5 seeds)

## B.2 VALIDATING DESIGN CHOICES FOR VALUE-BASED AGILE

We conducted an exhaustive search on the architecture of value-based AGILE from two different perspectives; (a) Hyper-parameters and (b) Architectures. Note that the same hyper-parameter search procedure of this section was applied to all other methods, and the same trend of the improvement in AGILE was found for other methods. In this section, AGILE used in the main results (Fig. 4, 5, 7) is called *AGILE-Tuned*. The barebone version of *AGILE-Tuned* is called *AGILE-Untuned*. We illustrate how each change contributes to an improvement in performance.

### B.2.1 HYPER-PARAMETER SEARCH IN AGILE

- **AGILE-Tuned without sync-freq-change:** In Mnih et al. (2015), the authors used the periodic syncing between the target and the main networks to alleviate the issue of frequently moving Q-value targets. In this work, we compare two extreme cases of the sync frequency: 10 depicted by *Sync-freq=10* in Fig. 13 (a) and 500 depicted by *AGILE-Tuned*.

- **AGILE-Tuned without graph-dim-change:** To understand the difficulty in expressing the action relations through a compact representation, we compare two hidden dimension sizes. The node-features are encoded in 32 (*Graph-dim=32*) or 64(*AGILE-Tuned*) dimensions.

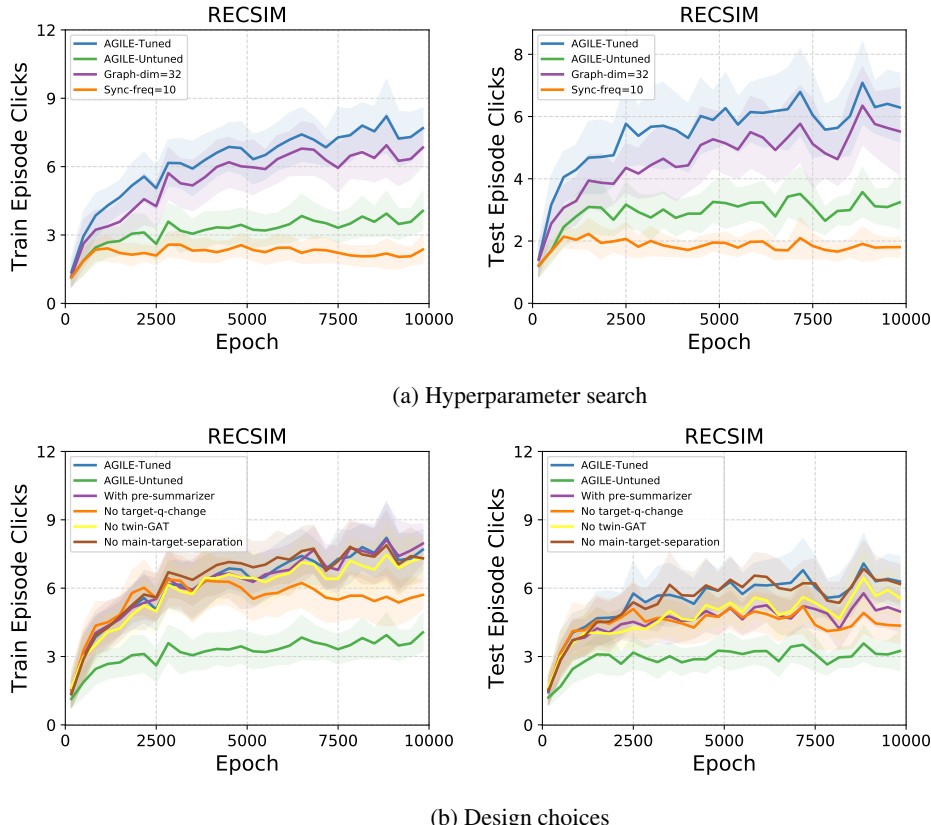

(a) Hyperparameter search

(b) Design choices

Figure 13: Results of value-based AGILE on RecSim CPR task (a) hyperparameter testing and (b) validating architecture design choices, on train actions (left) and test actions (right).

Figure 13(a) shows the result on both train and test actions. *AGILE-Tuned* outperformed all the methods. *Graph-dim=32* is slightly worse than *AGILE-Tuned* and *Sync-freq=10* fails to learn anything meaningful. Thus, frequently moving target network harms the agent's performance while sufficient expressiveness in the action graph improves the performance.

### B.2.2 DESIGN CHOICES OF AGILE

- **AGILE-Tuned with pre-summarizer:** In *AGILE-Tuned*, the concatenation of an action representation and the state is a node feature input to the GAT (Sec.C.3.1)). Here, we experiment and observe that adding a 2-layer MLP with ReLU over the node features does not help performance.

- **AGILE-Tuned without target-q-change:** Chen et al. (2019a) compute the target q-values for training CDQN using the list-action at the next time-step. But, there is another potential target q-value from the next item in the current list-action. Here, we compare these two methods to get the target q-value: (a) intra-list(*AGILE-Tuned*): the target q-value is from the next list index. (b) across-list (*No target-q change*): the target q-value is from the next timestep.

- **AGILE-Tuned without twinGAT-change:** In AGILE, the GAT output provides the utility network with the relational action representation and the action summary. Here, we compared two options: (a) Sharing GAT (*No twin-GAT*): there is a single GAT working to provide both of them. (b) Non-sharing GAT (*AGILE-Tuned*): this employs two different GATs for each.

- **AGILE-Tuned without main/target-encoder-separation-change:** In the implementation of AGILE, we compared two different architectural decisions. (a) *No main-target separation*: Separate the list-action encoder in CDQN for main and target Q-networks. (b) *AGILE-Tuned:* share the same list-action encoder for main and target Q-networks.

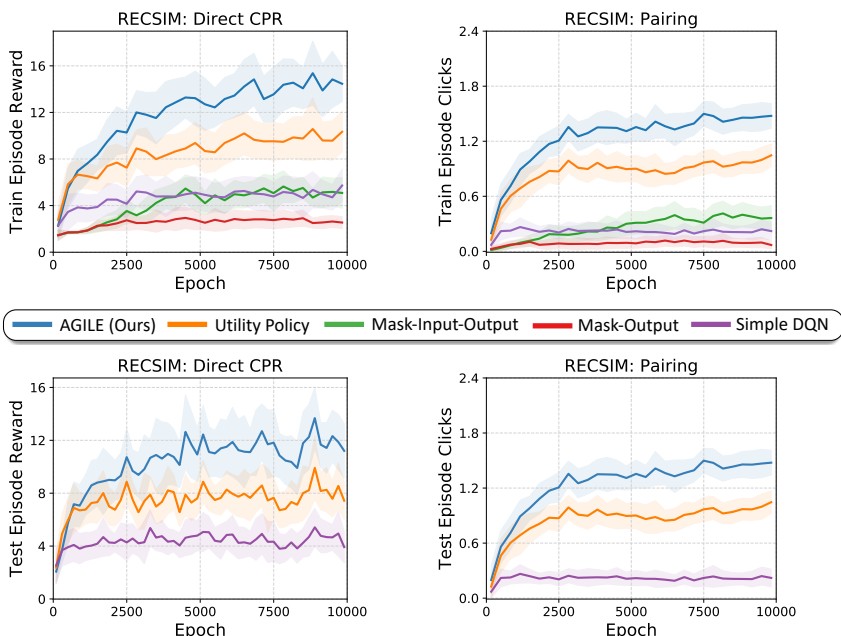

Figure 14: Comparison against the baselines on train (top) and test (bottom) actions on the Direct CPR (left) and Pairing (right) RecSim environments. Along with Figure 4, these results exhibit the same trend that AGILE consistently outperforms all the baselines on both train and test actions.

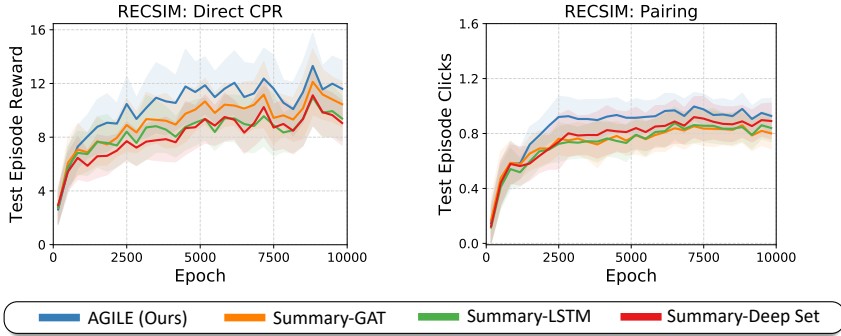

Figure 15: Comparison against ablations on the Direct CPR (left) and Pairing (right) RecSim environments. Along with Figure 5, these results exhibit the same trend that AGILE slightly outperforms the various summary ablations, which do not explicitly utilize relational action features and rely only on the summary vector to encode all the necessary action relations.

Figure 13(b) shows the result on both train and test actions. *AGILE-Tuned, With pre-summrizer*, and *No twin-GAT* showed the similar performance which is better than *No target-q-change*. The difference between *AGILE-Tuned* and *No target-q-change* is that that the cascaded network in *AGILE-Tuned* uses the target q-value from the intermediate list constructed. This is a more accurate target q-value as compared to the target q-value from another list from a future time-step.

## B.3    EFFECT OF DIRECT V/S INDIRECT REWARD IN RECSIM

In Fig.4, we studied the learning capability of AGILE under the action interdependence on RecSim in which the CPR itself was not directly visible to the agents. Therefore, agents need to indirectly understand how well the list-action is through user feedback (i.e., clicks). In this section, we strengthen our results by examining the consistency of results. The CPR is made directly visible to all the agents in this setting. So, we implemented an additional environment in RecSim where the reward is a sum of the click reward and the CPR metric. We call this environment as *Direct CPR*. Figure.14 shows the comparison of AGILE against the same baseline agents as in Fig.4 on the Direct

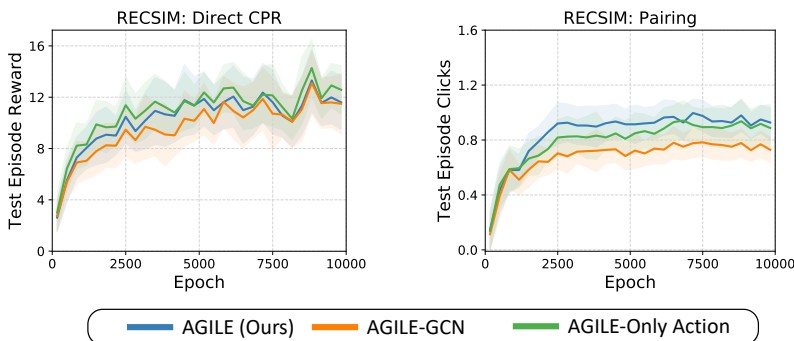

Figure 16: Analyses of (i) GAT v/s GCN and (ii) state-action graph v/s action-only graphs on (left) RecSim: Direct CPR environment and (right) RecSim: Pairing environment.

CPR RecSim. And in Fig.15 and Fig.16, we got the consistent result that AGILE slightly outperforms the ablations. AGILE outperforming the baselines shows that the knowledge of dependence on other available actions is crucial for optimal policy. This result is consistent with the indirect CPR optimization setting in Fig.4. So AGILE can perform well in the direct maximization of CPR.

### B.4 RecSim-Pairing Environment

In the experiment of Fig. 5, we found that in RecSim, the relation of items is easy to model such that AGILE could not outperform the ablations. In contrast, AGILE outperformed the ablations in CREATE and Grid World by correctly utilizing the action relation in decision-making. We hypothesize that these environments require complex relations between actions (e.g., tools and activators in CREATE). To this end, we implement the pre-defined pairings among items in RecSim such that clicks can only happen when the correct pairs of items are recommended. Since action relations are complex, AGILE is expected to outperform the ablations. Figure 14 shows that AGILE beats the baselines and in Fig.15 AGILE slightly but consistently outperforms the ablations. In Fig.16, AGILE outperforming AGILE-GCN shows that a GAT is capable of modeling the action relations correctly. AGILE converges faster than AGILE Only-Action. This shows that the state and the partially constructed list are crucial to learning to attend the other half in pairing items efficiently.

## C Approach and Baseline Details

### C.1 Details of Baselines and Ablations

Like AGILE, all baselines and ablations receive the state and action representations as input and output a Q-value or a probability distribution over available actions. Figure 17 describes the architectures of all the methods. Here, we discuss how each baseline and ablation is different from AGILE and why AGILE is expected to outperform them:

- **Mask-Output** (No representations, No input action set): This baseline assumes a fixed-action space that is known in advance. Since it does not use action representations, it cannot generalize to unseen actions or exploit the structure in action space. Moreover, it does not take the available action set as input and thus cannot solve tasks where action decisions require knowledge of other actions.

- **Mask-Input-Output** (No representations): By augmenting a binary availability mask of given actions to the state input of *Mask-Output*, this method can utilize the information about the available action set as a set. However, the input availability-mask is fed into an MLP, which lacks the inductive bias of order invariance of action set and cannot learn relations explicitly like graph networks.

- **Utility-Policy** (No input action set): By using action representations, this method can use the structure of action space for efficient training and also generalize to unseen actions. However, like *Mask-Output*, it does not utilize the available action set as part of the state and makes each action

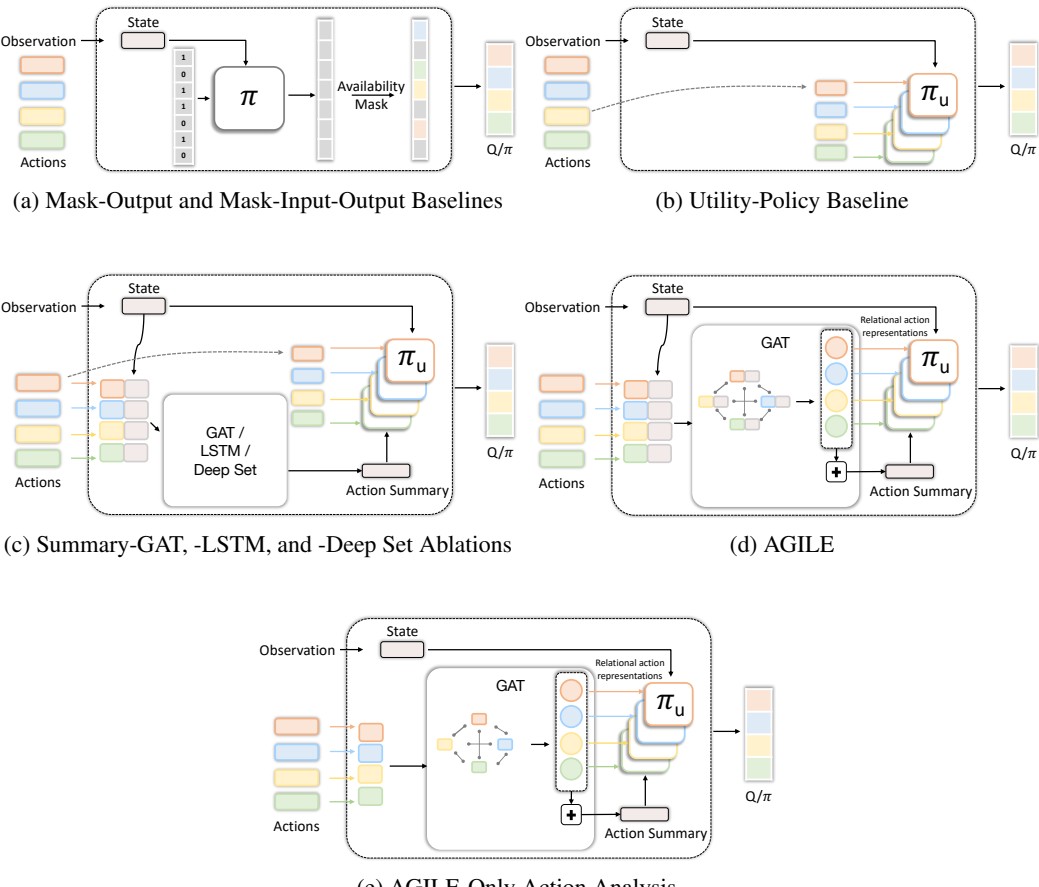

Figure 17: Architectures of all methods used as baselines (Sec. 6.1.1), ablations (Sec. 6.1.2), and analyses (Sec. 6.3.2). (a) Following prior work in SAS-MDPs (Boutilier et al., 2018; Chandak et al., 2020a) and invalid action masking (Huang & Ontañón, 2020; Ye et al., 2020; Kanervisto et al., 2020), the **mask-output** baseline masks out the unavailable actions assuming a known action set. The **mask-input-output** baseline additionally augments the state with the action availability mask. (b) Utility-policy (Jain et al., 2020) can generalize over actions by using action representations. However, it computes each action utility independent of other available actions. (c) **Summary** Ablations augment the utility-policy with an extra action-summary input, which is a compressed version of the list of available action representations. This compression can be done by mean-pooling over a *Bi-LSTM* output layer, or a *deep set*, or a graph network (*GAT*) processed node features. (d) **AGILE (Ours)** uses a GAT's node features both to compute an action set summary and as relational action features replacing the original action representations. While the action set summary is a compact representation of the available actions, it does not sufficiently scale when there are many actions or the task requires many action relations for each action decision. (e) **AGILE-Only Action** is the version of AGILE where the state is not used to compute action relations in GAT. This is a simpler architecture to learn but is not expected to work for certain tasks where action relations change depending on the state.

decision independent of other actions. Thus, it is expected to be suboptimal in all the tasks we consider.

- **Summary-LSTM**: This is an ablation where we do not utilize the per-action relational features computed by a GAT in AGILE. Instead, we use raw action features as input to the utility network. However, we utilize the available action set information by summarizing the set of input action representations into a vector. The summarization based on Bi-LSTM does not use the order invariance property of the action set, which makes it less efficient to learn than other summarizers. AGILE is expected to outperform all summary-only ablations in environments where several different action relations need to be modeled for computing different action utilities. This is true for complex environments like CREATE, where the agent needs to consider various tools and activators to make an optimal decision about its action choice. In such environments, learning a shared summary-vector for all action utility computations is inefficient and possibly prohibitive for learning. We note that the summary-ablations are sufficient in simple environments such as Grid World navigation. The agent just needs to summarize which of the two dig-lava skills are available. Such a summary is enough to compute the utility of each of the 7 available actions.

- **Summary-Deep Set**: This ablation utilizes the order invariance of action sets using a deep-set. While it still suffers from the limitations of lack of per-action relational action features, it is a low-parameter network and thus easy to train.

- **Summary-GAT**: This is a variation of AGILE, where raw action features replace the relational action features. Using a GAT to extract the action set summary exploits order invariance. However, being a fully-connected graph network, it is can be slow to train while not offering much more than the deep-set summarizer. This can be useful when certain action relations are predefined using domain knowledge (Section B.1).

- **Agile-Only-Action**: This variation of AGILE does not utilize state input in the GAT. Thus, the summary vector and relational action features are computed independently of state-context. While this is a simpler architecture to train, it can be insufficient in environments where the action relations vary depending on the state.

## C.2 DETAILS ON LISTWISE AGILE

AGILE is implemented based on the listwise RL architecture, CDQN (Chen et al., 2019a). As shown in Algorithm 1, CDQN builds a list-action incrementally by selecting N actions in sequence. For each list-index, a Q-network is used to select the best action. In addition to the usual state input, the partially built list-action is also an input to the Q-network. Concretely, the state and current list are encoded into latent embeddings for any list index. These are concatenated with each action representation in the remaining action set to build a list of nodes to be input into the action graph of AGILE. Treating it as a fully-connected graph, AGILE outputs a set of relational action features for all inputs actions. These are mean-pooled into a summary vector representing the entire action set. The representations of state, current-list, summary, and relational action features are used as input by a utility network (MLP) which outputs the Q-value of that particular action for the current list-index. The action with the maximum Q-value is added to the list, and the algorithm moves to the next list-index.

For training AGILE-based CDQN, we maintain a target Q-network that is synchronized periodically with the main Q-network. Suppose a tuple of $(s, a_{\text{list}}, r, \tilde{s})$ is sampled from the replay buffer. For list index $n$ between 1 and $N - 1$, we compute the target Q-value using the current state $s$ with partial list-action $a_{1:n}$ as input. However, for the last list index $n = N$, we compute the target Q-value using the next state $\tilde{s}$ for its first list index. A mean-squared error loss is used over all the list indices to train the main Q-network.

## C.3 NETWORK ARCHITECTURES

### C.3.1 ACTION GRAPH

The action graph takes as input the action representations and the state-information (we also include the list-embedding for Listwise AGILE; See Sec. C.2). Given the concatenation of the input components above, an optional 2-layer MLP with ReLU, called *pre-summarizer-mlp*, transforms it into

---

**Algorithm 1** Cascaded DQN: Listwise Action RL

---

1: **def** listwise_action():
2:     **Parameters**: Q-network $\phi$ - Encoders, GAT and Utility network
3:     **Inputs**: State $s$, actions $\mathcal{A}$, representations $\mathcal{C} = \{c_{a_0}, ..., c_{a_k}\}$, list-action length $N$
4:     **Initialize**: List Action $a_{\text{list}} = []$. Candidate Actions $\mathcal{A}' = \mathcal{A}$
5:     **for** $n = 1, \ldots, N$ **do**:
6:         Encode State: $e_s = \phi_{\text{s}}(s)$
7:         Encode Current List: $e_{\text{list}} = \phi_{\text{list}}(a_{\text{list}})$
8:         Build Graph Nodes: $\mathcal{V} = \{[e_s, e_{\text{list}}, c_a] \; : \; a \in \mathcal{A}'\}$
9:         Build fully-connected Adjacency Matrix of size $|\mathcal{V}|$: $\mathcal{E}$
10:       Relational Action Features: $e_{\text{relational}}^a = \phi_{\text{GAT}}(\mathcal{V}, \mathcal{E}) \;\; \forall a \in \mathcal{A}'$
11:       Action Set Summary: $e_{\text{summary}} = \frac{1}{|\mathcal{V}|} \sum_a e_{\text{relational}}^a$
12:       Q-values: $Q_n(s, a, \mathcal{A}') = \phi_{\text{utility}}(e_s, e_{\text{list}}, e_{\text{summary}}, e_{\text{relational}}^a)$    $\triangleright$ Ablations: $e_{\text{relational}}^a = c_a$
13:       Select action: $a_n = \arg\max_{a \in \mathcal{A}'} Q_n(s, a, \mathcal{A}')$
14:       Update: $a_{\text{list}} = a_{\text{list}} \cup a_n$. $\mathcal{A}' = \mathcal{A}' \setminus a_n$
15: **def** listwise_update():
16:     **Parameters**: Q-network $\phi$, Target Q-network $\phi_T$, discount factor $\gamma$
17:     **Inputs**: state $s$, action $a_{\text{list}} = a_1, \ldots a_N$, reward $r$, next state $\tilde{s}$
18:     **for** $n = 1, \ldots, N$ **do**:
19:         Use listwise_action($\phi$) to get Q-function: $q_n = Q_n(s, a_n, \mathcal{A} \setminus a_{1:n-1})$
20:         Use listwise_action($\phi_T$) to get target Q-value :

$$y_n = \begin{cases} r + \gamma \max_a Q_{n+1}(s, a, \mathcal{A} \setminus a_{1:n}) & n < N \\ r + \gamma \max_a Q_1(\tilde{s}, a, \mathcal{A}) & n = N \end{cases}$$

21:     Optimize Loss $\mathcal{L} = \sum_{n=1}^N (q_n - y_n)^2$

---

the node features for the action graph. Then, the resultant node features are passed on to two GAT layers followed by a residual connection. The same architecture of GAT is duplicated with different weights to provide separate pathways to compute the summary vector and relational action features. Therefore, we have two different sets of node features. The first set of node features is mean-pooled to produce the action-summary vector, which is put through a 2-layer MLP with ReLU to post-process before being passed on to the utility network. The other set of node features (i.e., relational action features) is directly fed into the utility network.

**AGILE with GCN**: When the GCN is used in the action graph, the same input as above goes into a linear layer to compress the size. Subsequently, the resultant node-features are used in GCN message-passing based on the normalized adjacency matrix, with all the diagonal elements made $0$. Otherwise, the GCN reduces to learning the same edge coefficients for all the nodes. The rest of the following architecture is the same as GAT above.

**Summary-GAT**: In this ablation, instead of the relational action features being fed into the utility network, the raw action representations are used. The rest of the architecture remains the same.

### C.3.2 Summarizers: Bi-LSTM and Deep Set

**Bi-LSTM**: The raw action representations of candidate actions are passed on to the 2-layer MLP followed by ReLU. Then, the output of the MLP is processed by a 2-layer bidirectional LSTM (Huang et al., 2015). Another 2-layer MLP follows this to create the action set summary to be used in the following utility network.

**DeepSet**: Similar to the Bi-LSTM variant of the summarizer, we employed the 2-layer MLP with ReLU followed by the mean pooling over all the candidate actions to compress the information. Finally, the 2-layer MLP with ReLU provides the resultant action summary with the following utility network described in the next subsection.

### C.3.3 UTILITY NETWORK

We implemented two types of utility networks to show the potential of AGILE architecture working with different kinds of RL algorithms and environments. Both the utility networks take as input the same components: the state-information, the relational or raw action representations, and the action summary of the action graph (See Sec C.3.1). The utility network is a 2-layer MLP applied parallelly on all these inputs corresponding to each action. It computes a scalar value for each action.

**Policy Gradient(PPO)**: The action utility values are used as logits of a Categorical distribution, which is then used to train the AGILE architecture with policy gradient algorithm, PPO.

**Value-based(CDQN)**: *List Encoder:* An intermediate list that contains the currently selected list items at each intra-list time-step is passed on to a single layer gated recurrent network (GRU (Cho et al., 2014)) followed by a 2-layer MLP to extract the compact representation of the intermediate list. See Algorithm 1 for details on this.

*Q-network:* As in Sec C.3.1, the input components for the utility network are the raw action representation, the state-information, the list-embedding, the node-features, and the action set summary from the action graph — $(s_t, e_{\text{list}}, e_{\text{node}}, e_{\text{summary}}, a_k)$ in Algorithm 1. These are concatenated into a single vector and passed on to a 2-layer MLP with ReLU to compute the Q-value of an item.

### C.3.4 ACTION REPRESENTATION NETWORK

**Hierarchical VAE (CREATE)**: The CREATE action representations are borrowed from Jain et al. (2020) directly. The Hierarchical VAE network takes as input a list of behavioral trajectories representing each action (tool) and reconstructs it with a two-layer hierarchy of VAEs. We refer the reader to Appendix D.3.1 of Jain et al. (2020) for complete details of the Hierarchical VAE network.

**VAE in Deep and Wide architecture (Real World recommender system)**: In a VAE, the encoder is implemented with a 5-layer MLP with a Batch Normalization layer (Maas et al., 2013) and LeakyReLU (Maas et al., 2013). On the other hand, the decoder is implemented with a single MLP to process the input, followed by the same architecture of MLPs used in the encoder. Finally, a 2-layer MLP with Batch Norm and LeakyReLU and the hyperbolic tangent activation function is used to reconstruct the input instance.

### C.3.5 REWARD INFERENCE NETWORK (USER MODEL IN REAL WORLD RECSYS)

The user model takes as input the user information(i.e., a concatenation of user attributes and a sequence of the user interactions) and a set of item embeddings in the list. The user information is passed on to a single layer gated recurrent network(GRU (Cho et al., 2014)) followed by a 2-layer MLP to extract the compact representation of the state. The same GRU network architecture (with different weights) processes the set of item embeddings into a list-embedding. Finally, a 2-layer MLP takes as input the concatenation of those two embeddings(i.e., state-embedding and list-embedding) and provides the scores of items in the list followed by the sigmoid function to transform to the individual click likelihood.

## D EXPERIMENT DETAILS

### D.1 IMPLEMENTATION DETAILS

We used PyTorch (Paszke et al., 2019) for our implementation, and the experiments were primarily conducted on workstations with either NVIDIA GeForce RTX 2080 Ti, P40, or V100 GPUs on Naver Smart Machine Learning platform (NSML) (Kim et al., 2018). Each experiment seed takes about 4 hours for Grid Navigation, 60 hours for CREATE, 8 hours for RecSim, and 15 hours for Real-Data Recommender Systems, to converge. We use the Weights & Biases tool (Biewald, 2020) for logging and tracking experiments. All the environments were developed using the OpenAI Gym interface (Brockman et al., 2016). For training Grid Navigation and CREATE environments, we use the PPO (Schulman et al., 2017) implementation based on Kostrikov (2018). For the recommender system environments, we use DQN (Mnih et al., 2015). We use the Adam optimizer (Kingma & Ba, 2014) throughout. We attach the code with details to reproduce all the experiments, except the real-data recommender system.

## D.2 HYPERPARAMETERS

We build on hyperparameters used in prior work on CDQN (Chen et al., 2019a) and utility policy (Jain et al., 2020). The hyperparameters for the additional components introduced in AGILE, baselines, and ablations are shown in Table 3. The environment-specific and RL algorithm hyperparameters are described in Table 4.

| Hyperparameter | Value |
| --- | --- |
| **AGILE** | |
| number of GAT layers | 2 |
| number of attention heads | 1 |
| number of message passing steps | 1 |
| leakyReLU alpha | 0.2 |
| graph hidden dimension | 64 |
| Residual Connection | True |
| **Mask-Input-Output** | |
| Availability mask MLP hidden size | 64 |
| **Ablations** | |
| Summary-LSTM hidden size | 64 |
| Summary-LSTM directions | 2 |
| Summary-LSTM number of layers | 2 |
| Summary-Deep-Set hidden size | 64 |
| Summary-GAT hidden size | 64 |
| Summarizer pooling operation | mean |

Table 3: AGILE, baseline and ablations: Hyperparameters for additional components

## D.3 HYPERPARAMETER TUNING

Initial hyperparameters are inherited from the prior works (PPO: Jain et al. (2020); CDQN: Chen et al. (2019a)). To ensure fairness across all baselines and our methods, we use the original hyperparameters for the RL algorithms, such as learning rate, entropy coefficient, etc. (Table 4. We choose a sufficiently large number for total epochs (DQN) and total environment steps (PPO) to ensure all the methods converge.

Our main contribution is the AGILE architecture. Thus, to correctly validate against baseline and ablation architectures, we search over shared parameters together and generally observe the same trend across all methods. This was expected because of the shared underlying implementation of the RL algorithm. Specifically, we searched over the following shared parameters, usually over 5 seeds and sometimes 3 seeds:

- **Hidden dimension in DQN networks**: We searched over $\{32, 64, 128\}$ and found that $64 \approx 128 > 32$. We choose 64 as the base hidden dimension across all encoders, summarizers, and GATs.

- **Target network sync frequency**: This was a sensitive parameter for DQN training. We searched over $\{10, 20, 100, 500, 1000\}$ and found 500 to be the best performing across all methods.

- **Weight sharing in CDQN**: In Chen et al. (2019a), the authors did not share the weights of Q-nets in their Cascading architecture, but we empirically observed that sharing the weights of them improved the performance. This is likely due to the nature of RecSim tasks requiring similar items to be recommended across the slate to maximize CPR.

- **Training batch size (DQN)**: We searched over $\{32, 64, 128, 256\}$ on RecSim and chose 128 as, beyond that, we observed diminishing returns. For Real RecSys, we used the same batch size.

- **Training batch size (PPO)**: We searched over $\{4096, 8192\}$ for Grid World and $\{3072, 4608\}$ for CREATE and observed that larger batch sizes are better. We did not increase the batch size further due to limits on GPU memory size.

| Hyperparameter | Grid world | CREATE | RecSim | Real RecSys |
|---|---|---|---|---|
| **Environment** | | | | |
| action representation size | 11 | 134 | 20 | 32 |
| observation space | 162 | $84 \times 84 \times 3$ | 10 | 507 |
| candidate actions per episode | 7 | 25 | 20 | 20 |
| max. episode length | 50 | 30 | 15 | 10 |
| **RL Training** | | | | |
| training batch size | 8192 | 4608 | 128 | 128 |
| parallel processes | 64 | 48 | 16 | 16 |
| discount factor | 0.99 | 0.99 | 0.99 | 0.99 |
| learning rate | 1e-3 | 1e-3 | 1e-4 | 1e-4 |
| hidden layer size | 64 | 64 | 64 | 64 |
| **PPO** | | | | |
| entropy coefficient | 0.05 | 0.005 | | |
| continuous action entropy coefficient | - | 0.0005 | | |
| total environment steps | $8 \times 10^6$ | $10^8$ | | |
| value loss coefficient | 0.5 | 0.5 | | |
| PPO epochs | 4 | 4 | | |
| PPO clip parameter | 0.1 | 0.1 | | |
| **DQN** | | | | |
| epochs | | | 10000 | 5000 |
| target network sync frequency | | | 500 | 500 |
| Q-network hidden dimension | | | $256 \times 64$ | $256 \times 64$ |
| list encoder dimension | | | 64 | 64 |
| action encoder dimension | | | 32 | 32 |
| list size | | | 6 | 6 |
| replay buffer size | | | $10^6$ | $10^6$ |
| initialize replay buffer size | | | 5000 | 100 |
| epsilon decay | | | $1 \rightarrow 0.1$ | $1 \rightarrow 0.1$ |
| epsilon decay last epoch | | | 500 | 250 |

Table 4: Environment/Policy-specific Hyperparameters

Below, we detail the hyperparameters and network variations searched specifically for the graph attention network (GAT) used in AGILE and its variants.

- **Skip Connection**: Adding skip connection was crucial to making AGILE learn well.

- **Number of graph attention layers**: We searched over $\{1, 2, 3\}$ layers and found 2 layers to be the best performing for both PPO and CDQN based policies.

- **Number of attention heads**: We compared having $\{1, 2\}$ attention heads and found no meaningful learning for 2 attention heads.

- **Number of message passing steps**: Adding more message-passing steps than 1 did not improve AGILE performance.

Other relevant design choices are described and validated in Section B.2.

