# OpenReview forum: "Know Your Action Set: Learning Action Relations for Reinforcement Learning"
_ICLR.cc/2022/Conference — ICLR 2022 Poster_

### Official Review · Reviewer_mRLy · 2021-11-01

**Correctness:** 3
**Technical Novelty And Significance:** 3
**Empirical Novelty And Significance:** 3
**Recommendation:** 8
**Confidence:** 5

**Main Review:**

Strengths

The motivation is solid and intuitive, the summaries of related work are insightful, the environments involve variable sets of actions for which the pairwise relations are clearly important, and the discussion of experimental results is enlightening. AGILE’s benefits are most clear for the CREATE environment.

Weaknesses

AGILE’s margin of improvement over baselines is not large enough to outweigh the spurious effects of inadequate hyperparameter tuning (a well-known issue in deep RL). Unfortunately, the paper makes no mention of hyperparameter tuning. GAT hyperparameters are listed in Table 1, but hyperparameters for baseline architectures (like LSTM and Deep Set) are missing, and standard learning hyperparameters (like learning rate) are not mentioned at all. We are left to wonder whether AGILE’s moderate gains would be maintained if the hyperparameters had been more thoroughly tuned for the baselines. This seriously undercuts the conclusions we can draw from the experimental results.


**Summary Of The Paper:**

This paper proposes an architecture (AGILE) for handling variable action sets through pairwise attention between the currently available actions, where the action set is free to change between episodes, and each action is described by an associated feature vector. In evaluations against baseline architectures and ablations on a diverse set of suitable environments, AGILE performs the best overall.

**Summary Of The Review:**

This work has much to admire, and could make an important contribution if the uncertainty regarding its hyperparameter tuning didn’t cast such doubt on the experimental results.

--- POST-REBUTTAL UPDATE ---

The authors have provided much more information about their hyperparameter tuning, and have performed additional experiments which provide additional insights. These changes address my concerns, and I have revised my evaluation.

---

> ### Author Response · Authors · 2021-11-23
> **Added hyperparameter-tuning details and more experiments to verify experimental correctness**
>
> We thank you for your constructive feedback. We appreciate your positive comments on motivation, related work, environments, and experiment discussion. We agree with the criticism on insufficient details about hyperparameters and their tuning. We address this in the following way:
>
> - **Response**
> We adopt the RL relevant hyperparameters (such as learning rate, entropy coefficient) from prior work and primarily tune the parameters specific to the added components (i.e. GAT, summarizers) for fairness. Specifically, we adopted and fixed all the crucial hyperparameters from the original respective papers (Chen et al., 2019a and Jain. et al (2020)). Then, the mask-baselines (mask-input and mask-output) and AGILE variants are implemented on top of the original architectures and hyperparameters. Certain RL parameters affected all the methods in the same way - e.g. target network sync frequency and batch size in DQN, PPO. The same trend was expected because all methods share the same underlying RL algorithm, and we validated this experimentally. We mainly tuned the hyperparameters of the additional components in GAT, namely hidden dimension, skip connection, graph attention layers, attention heads, and message passing steps, as efficiently training GAT with RL reward is challenging.
>
> - **Paper Changes**
> We add a complete hyperparameter list in Appendix D.2 and the procedure for hyperparameter tuning in Appendix D.3 for all environments and methods. We also add the description of network architectures used in AGILE and ablations to Appendix C.3. We experimentally validate various design choices for the RecSim environment in B.2.
>
> - **Experiment Changes**
> 1. Based on your feedback, we revisited the hyperparameter tuning for all the methods (baselines and ablations) to ensure the evaluation is fair. We went a step further and tested various architectural changes of AGILE and its variants. We found a few important design choices that improved the performance of all CDQN-based AGILE methods, i.e., on the recommender system environments. These include slowing target network sync frequency, removing the MLP before GNNs or summarizers, and increasing hidden dimensions to 64. We add these experimental comparisons in B.2
>
> 2. To further validate the correctness of experimental results (and not due to spurious correlations), we added two new RecSim tasks. Thus, in total, we evaluate on **3 separate RecSim tasks** - (i) Indirect-CPR (Original env): User Model depends on CPR but reward reflects this indirectly via stochastic clicks, (ii) Direct-CPR (Appendix B.3): User model depends on CPR and reward = click + CPR-reward. So, direct feedback. (iii) Pairing (Appendix B.4): Added a special item for each category (like activators in CREATE). Clicks can only happen when the correct pairs of items are recommended.
> We observe consistent trends in all environments and on all RecSim tasks:
>
> - **AGILE > baselines**, always.
> - **AGILE >= ablations** and **AGILE >= AGILE-GCN**, especially when the task requires learning many action-specific relations which cannot be modeled by a single summary vector easily (e.g. pairing-RecSim and CREATE).
> - **AGILE >= AGILE-only-action**. AGILE-only-action is optimal on all RecSim tasks because action relations are predefined, based on item categories and thus do not vary when state changes. AGILE-only-action underperforms in Grid world and CREATE.
>
> Finally, we hope to answer your concern about **the improvement over baselines not being large enough**. We highlight below why we believe these gains over baselines are significant (Fig. 4):
> 1. Grid World: the difference between AGILE and the utility-policy has an interpretable meaning. AGILE learns the maximum reward possible as it uses different strategies for different availability of dig-lava skills (see qualitative results in Fig. 6(b)). Whereas, utility-policy’s reward is the maximum reward possible for the safest strategy of going around both the lava columns - which is the best a policy can do without knowing what all the available skills are. This, in fact, confirms that the baselines are well-tuned and are working up to their potential.
> 2. CREATE: we observe a performance gain of about 2x for both training (seen) and test (unseen) actions between AGILE and the next best baseline (utility policy).
> 3. For RecSim, the performance gain is 35%. Since CPR is only one component of user choice, the improvement expected from good CPR optimization is not too high in magnitude. Crucially, we observe the same trend across 3 RecSim tasks averaged over 5 seeds, which means the results are not due to randomness.
>
> **References**
> [1] Jain, Ayush, et al. "Generalization to new actions in reinforcement learning." ICML (2020).
> [2] Chen, Xinshi, et al. "Generative adversarial user model for reinforcement learning based recommendation system." ICML (2019)

---

### Official Review · Reviewer_3F2U · 2021-11-02

**Correctness:** 3
**Technical Novelty And Significance:** 3
**Empirical Novelty And Significance:** 3
**Recommendation:** 6
**Confidence:** 3

**Main Review:**

The central point of the paper is that having variable action spaces requires learning interdependence relations between possible actions. This point is illustrated well in the paper; AGILE outperforms benchmarks that ignore action interdependence (in particular, Jain et al.) and the qualitative results in Section 6.2 show that attentions learned by the GAT are reasonable.

My main concern with the paper is that a lot of important details are missing in the exposition. In particular, Section 4 relies too much on previous work in order to explain the architecture in Figure 2 without contextualizing that previous work well within the problem setting of AGILE. For instance,
* How are the nodes of the GAT determined? As far as I understood, when a finite set of base actions are given, each node corresponds to one of the actions. Then, what happens when a set of base actions is not known (which is certainly a setting AGILE aims to address)?
* In the case of listwise RL, I understand how the number of possible lists is combinatorial and this is problematic for AGILE. However, I do not understand how this issue was resolved; the only explanation given is that "the lists were built incrementally, one action at a time." What does this precisely mean?
* In Section 5.3.2, the authors mention that "they collected interaction data" for the real-world recommender system environment. As far as I understood, this data appears for the first time in this paper however there is not enough details regarding how the data was collected, how it is structured, how the reward models were trained, etc.
* I think how the experiments with train vs. test actions are conducted need to be clearer. As far as I understood, each model was trained using train actions only but for the bottom results in Figure 4, they were evaluated when test actions were also available. Then, how are the actions in CREATE are represented? I assume not all actions are one-hot encoded in this environment; otherwise, how would it be possible to generalize to actions that are never seen before?

Some minor comments:
* How baselines in Section 6.1.1 were introduced is very helpful in understanding the related work. It is very clear there in what aspects AGILE differ from the existing work.
* Needing a hammer to be able to use a nail is used as the main example of how action interdependences come into play. At first, I was confused about how attentions in a graph is able to capture a directional dependence like this but Section 6.2 explains it perfectly. Considering how central the nail-hammer example is to the exposition, maybe how attention weights relate to it can be explained even earlier.







**Summary Of The Paper:**

This paper considers the reinforcement learning problem where the action space is fixed. Having a variable action space makes it necessary to learn interdependence between different actions (use of some actions might depend on the existence of other complementary actions), which is modelled via a graph attention network.

**Summary Of The Review:**

The key idea of the paper is illustrated well but many details regarding the approach and the experiments are left out.

---

> ### Author Response · Authors · 2021-11-23
> **Added necessary details on methods and experiments**
>
> We thank you for your meaningful feedback. We appreciate your comments on the central idea being clear and well-illustrated, especially using qualitative results. For your main concern about lack of certain important details, we have made the following modifications:
>
> ### Method Details
> - **How are the nodes of the GAT in AGILE determined?**
> Referring to the problem formulation (Sec 3), the base set of actions corresponds to the entire set of actions that can be sampled from. This is unknown to the agent and it only receives a candidate action set of size K from the base actions, in the form of K action representations. Thus, our problem formulation assumes an extra input to the agent as compared to conventional RL - a list of available actions’ representations. *This extra input determines the nodes of the GAT in AGILE and other methods*. Figure 17 (Appendix C) gives a visual insight into how different methods utilize these action representations as input.
> - **How is listwise RL’s combinatorial action space dealt with?**
> We follow Chen et al., 2019a for CDQN architecture to deal with listwise RL. We have added a detailed description and algorithm to Appendix C.1. Concretely, CDQN builds a list-action incrementally by selecting N actions in sequence using a Q-network. For each list-index, the Q-network selects the best action by computing all values. Along with the state, the currently built list is also an input to the Q-network.
> - **Explain how attention weights relate to the dependence in the nail-hammer example.**
> Thanks for the suggestion. We have added this in the caption of Figure 1 and Section 4.1. In a learned GAT, the attention weights are expected to be high for closely interdependent actions, such as nail & hammer and tape & hook.
>
> ### Experiment Details
> - **Need more details on the data collection for real-world recommender system**
> Appendix A.4 details the data collection and RL environment description of the real-data RecSys. Based on your feedback, we have structured it in a form that improves clarity. To answer your questions, the user-interaction data is collected from two different periods. User models (i.e. reward models) are trained on both datasets separately, one is used for training and the other for evaluation. Similarly, the action representations were trained on the training dataset by encoding raw item attributes (reward points, text description, etc.).
> - **How does learning with train actions generalize to unseen test actions? How are the actions represented?**
> Generalization to unseen actions is possible due to continuous action representations as originally shown in Jain et al (2020). These representations must be obtained separately from the RL agent. Specifically, for RecSim, we predefined action representations based on a Gaussian mixture model distribution and then split them, CREATE uses learned action representations that encode tool behavior in the form of trajectories (thus given new tool’s trajectories, it can get representation), and similarly, Real Recsys learns action representations from raw item attributes. Complete descriptions of all action representations are added to Appendix A.

---

### Official Review · Reviewer_FMiS · 2021-11-03

**Correctness:** 3
**Technical Novelty And Significance:** 3
**Empirical Novelty And Significance:** 3
**Recommendation:** 8
**Confidence:** 4

**Details Of Ethics Concerns:**

This paper does not pose any direct ethics concern for the best of my knowledge

**Main Review:**

Strengths:

Learning action relations and dependence is an important problem in Rl, especially when the action set is varying according to the environment's constraints, as this is often the case in real-world scenarios and applications. Authors address a subset of this broader problem, namely when the action set only changes at the beginning of a task instance.

The main contribution and the novelty of this paper lie in the utilization of graph attention networks to learn the dependencies of actions using a fully connected action graph. The formulation of the fully connected action graph and the graph attention network is sound, though there are some questions regarding graph formulation which I ask below.

The authors demonstrate the generalizability of the policy architecture through training both value-based policy-based RL agents. Specific implementation details of how AGILE can be used in widely used RL methods (PPO, DQN) are also given.

Another strength of the paper lies in its evaluation, where experiments are done both in benchmarks and in applications, the latter with simulated and real-world data. The selection of benchmark environments is suitable for the evaluation. I particularly like the inclusion of the application based evaluation. There are some concerns with the method selection for evaluation which I share below.

The authors also give further insights into how the attention of the graph network is performed by qualitatively examining the method’s performance in different domains that were evaluated. I encourage the authors to have a more in-depth discussion on this as in its current form it is sparse. For example, it would be useful for the reader if contrastive cases are given for the presented examples.

The paper is also well written and easy to understand. The formulation of the architecture is clear, though some network descriptions and parameters are missing.

Weaknesses:

The main weakness I see in this paper is the lack of positioning the work relative to action varying RL methods. The authors do not compare the work of (Chandak et al, ,2020b) to the proposed method (though they have briefly discussed it in the related work section) computationally. While authors do use masked action sets and utility policy as baselines, other action varying baselines are needed for a comprehensive evaluation.

Another minor weakness in the paper is the lack of clarity in the action graph formulation. Authors mention that domain knowledge can be used if action relations are predefined and known beforehand, but does not give much detail on how and to what extent this can affect the performance.


----------------------------------------------------------------------

After the rebuttal

I appreciate the clarifications made about the selection of the benchmarks and I agree with the authors here. I especially commend authors for doing an in depth qualitative analysis and adding contrastive cases, this makes a stronger case for the paper's contribution. I have updated my score accordingly.


**Summary Of The Paper:**

This paper introduces a novel policy architecture (AGILE) for RL agents that learn action interdependence from a varying action space. A graph attention network is used to calculate the action utility and to summarize the action set input. Authors argue that this architecture allows the RL agents’ to learn action relations that lead to optimal behaviour in a changing action space environment. This architecture is then evaluated in three benchmark domains. Further evaluations are done in the recommender systems context, with both simulated systems and using real-world data. Results seem to indicate that the proposed method outperforms non-relational RL methods in most cases.

**Summary Of The Review:**

This paper presents an approach that addresses an important problem in RL, namely how to act optimally in an environment with varying actions. The proposed method seems to perform well against the compared methods. Though more baselines are needed, especially from closely related methods to draw conclusions. Evaluation is done using a selection of benchmarks and application domains which I view as a positive. The architecture is described in sufficient detail, though some sections need further clarifications.


[1] Huang, S., & Ontañón, S. (2020). A closer look at invalid action masking in policy gradient algorithms. arXiv preprint arXiv:2006.14171.

---

> ### Author Response · Authors · 2021-11-23
> **Added discussion on related work, contrastive qualitative examples, and domain knowledge experiment**
>
> We thank you for your detailed feedback. We appreciate your positive comments on the problem formulation, use of GATs to learn action dependence, flexibility with value-based and policy-based methods, environments, evaluation, and writing. We address your recommendations about related work, richer qualitative analysis, and better clarity of action graph architecture below:
>
>
> ### Related work where action space changes
> There are prior works where the action space keeps expanding in lifelong learning (Chandak et al, 2020b) or where certain actions become invalid, like in games (Huang et. al, 2020). In contrast, our proposed “varying action space RL” setting requires learning in a constantly varying action space where actions can be removed, new actions can be added, and the action set can be completely replaced in an episode.
> 1. **Chandak et al, 2020b** is thus not a valid baseline, as their method is about how to adapt better when new actions are occasionally added to an existing action set during lifelong learning. It cannot accommodate pre-specified action representations which are necessary for generalization and does not have provisions for removing or replacing actions. We have added this discussion to Sec 2.2
> 2. **Huang et. al, 2020**: Thank you for pointing out this relevant work. Their proposition is that when certain actions become invalid, they must be masked out from the output action probability distribution. This is exactly the *Mask-Output* baseline described in Sec 6.1.1 (added citation) and Appendix Figure 17(a). This is not a coincidence, since the problem of dealing with invalid actions (Huang et al, 2020, Ye et al., 2020 and Kanervisto et al.,2020) is an example of Stochastic Action Sets-MDPs (SAS-MDPs), which we already compare against. We reiterate that these methods are unsuitable for our problem setting as they cannot deal with unseen actions, but are still compared as baselines on train actions. This discussion on invalid-action RL prior works is valuable and added to related work Sec 2.1.
>
> To the best of our knowledge, only the utility-policy baseline (Jain et. al, 2020) is capable of working under our varying action space RL setting (but suffers due to lack of learning action relations). Please let us know if you know any other prior works for baseline.
>
> ### Richer Qualitative Analysis: more in-depth discussion and contrastive cases
> Based on your feedback, we have improved Fig. 6 and added video results to the website to show how the baseline fails without the knowledge of available actions. It is unable to exploit the shortcut digging skills and always follows the same suboptimal path in Grid World. Likewise, it cannot identify the most common item category in RecSim, which is necessary to optimize CPR. We also add this discussion to Sec 6.2.
>
> ### Better Clarity of Action Graph: network descriptions and incorporating domain knowledge
> 1. To **improve clarity** of our methods and comparisons, we have added complete Network Descriptions to Appendix C.3, hyperparameter details to D.2-3, and a visual and descriptive comparison of AGILE against baselines and ablations to C.1.
> 2. Incorporating **domain knowledge**: We add an experiment to show that using domain knowledge to remove edges from the action graph improves learning speed and stability. (Sec 4.1 and Appendix B.1). In Grid World, we use the knowledge that since directional actions are always available, the only relevant action relations are the ones with the variable actions (2 out of 4 actions, including both dig-lava skills). So, while building the action graph, instead of having a fully-connected structure, we only keep connections to and from the 2 variable actions. The rest of the AGILE architecture and algorithm stays the same. This reduces the number of bidirectional edges from 7^2 = 49 to 29 by removing 20 edges between always available actions. As shown in Fig. 12, the learning speed of AGILE is slightly accelerated. We expect domain knowledge to help more when the action space is large and more complex action relations need to be learned by the GAT.
>
> **References**
> [1] Chandak Y. et al. Lifelong learning with a changing action set. In Proceedings of the AAAI Conference on Artificial Intelligence, 2020b.
> [2] Huang S. & Ontañón S.. A closer look at invalid action masking in policy gradientalgorithms.arXiv preprint arXiv:2006.14171, 2020.
> [3] Ye D. et al. Mastering complex control in moba games with deep reinforcement learning. In Proceedings of the AAAI Conference on Artificial Intelligence, volume 34, pp.6672–6679, 2020.
> [4] Kanervisto A. et al. Action space shaping in deep reinforcement learning. In 2020 IEEE Conference on Games (CoG), pp. 479–486. IEEE, 2020.

---

### Official Review · Reviewer_RWVF · 2021-11-03

**Correctness:** 4
**Technical Novelty And Significance:** 2
**Empirical Novelty And Significance:** 3
**Recommendation:** 8
**Confidence:** 3

**Main Review:**

This work frames itself as an extension of Jain et al. 2020 to consider the relations between actions rather than evaluating their utility independently. In this regard it is a moderately incremental development, so the bar for the quality of the study should be reasonably high.

The motivation for the work is sound and reasonably well-argued, and the positioning with respect to related work is quite clear.

The empirical study is interesting, and covers a good variety of benchmarks with quantitative as well as some qualitative analysis.

I find there is some room for improvement in the exposition of certain details, and in aligning the discussion more closely with the empirical findings.
My main concern is about the description of the ablations in S6.1.2 and S6.3.1, especially because some of the ablations outperform the proposed architecture, at least in the recommender systems experiments.
Summary-LSTM and Summary-DeepSet clearly use different architectures to produce the action set summary. But it’s unclear to me how they condition on the action. Do they use the raw action representations like Summary-GAT?
I’m also unsure about the authors’ assertion that Summary-GAT can clearly show the importance of using action interdependence: the utility function can still compare the action representation to the summary of available actions (which can, for example, use different parts of its representation to correspond to different specific actions). The specific definition of interdependence referred to here should be made more clear.
There is also no comparison of the total number of parameters or computation used by the different models, which makes it harder to understand the importance of the structural choices.
In S6.3.1, GCN is not defined (I assume this is Graph Convolutional Network, but there are many missing details).
I would also like more detail on the qualitative analysis: it’s unclear how the figures were produced. Do arrow widths correspond to attention weights? Are they left off the figure below a certain threshold? I also find Fig 6c (labelled 6b) quite hard to parse.

Clarifying details aside, I don’t feel the discussion aligns well with the full set of empirical results. The authors suggest that noisy/sparse rewards might make a GAT harder to train, but don’t show any particular evidence that this is the defining difference in the RecSim case. A potential followup would be to make the rewards sparser or noisier in the other tasks, to see if the GAT-based approach once again falls behind.
Overall, it feels that a more thorough discussion of the difference between the RecSim experiments and the other environments needs some more attention. One question might be how relevant the quality of the raw action representations is?
I also find the concluding remarks somewhat misleading. In S6.3.2 it says that Fig 7 shows a drop in performance across all environments when using AGILE-Only Action, but there is an increase in performance on RecSim, as far as I can see.
This claim is then echoed in the final sentence, that relational knowledge is crucial for optimal decision-making (while the ablations seem to suggest that relational knowledge is sometimes less important than affordance knowledge).

I would appreciate it if the authors could clarify the role of these ablations, and which conclusions may be drawn from them.

Minor comments:
 - The notation for the action set summary is a bit weird (S normally being reserved for states). Maybe \bar{c}^R would better indicate the average over action features?
 - The equation for the action set summary should have c^R rather than c?
 - Last sentence about Action Utility should say the score can be used as a logit fed into a softmax, rather than it can be used as a prob distribution
 - pi(a|s’) should also condition on c^R?
 - I appreciate the inclusion of some more anecdotal comments about which implementation decisions were important (residual connections, parameter sharing)

----------------------

The authors have comprehensively addressed my concerns, both by improving their algorithm's performance and reflecting appropriate changes in both the description and discussion of ablations (and qualitative analysis); I am raising my score as a result.


**Summary Of The Paper:**

This paper tackles an RL problem setting in which the actions available to an agent vary from episode to episode, and the optimal action in some states depends on the other actions that are available.
The authors’ approach to this setting is to use a graph neural network to process all available actions, both to summarise the available set and to produce a relationally-informed representation for each action. These are then fed, with the state, into a utility function to give an action value or logit.
Experiments in a number of benchmarks show the value of including information about the available actions.

**Summary Of The Review:**

The work is reasonably sound overall (if somewhat incremental) but is let down by the slightly unclear set of ablations and discussion of the empirical results. I would consider raising my score if the authors can compellingly clarify what conclusions can be drawn from their study.

----------------------

I believe the authors have addressed all of my concerns in their responses and updates to the paper.

---

> ### Author Response · Authors · 2021-11-23
> **Discussion on AGILE ablations and Conclusive RecSim results.**
>
> We thank you for your detailed feedback. We appreciate your positive comments on the motivation, benchmarks, and experiments. We address your questions about experimental results, description of ablation architectures and the details of the qualitative analysis below:
> ### Description and comparison of ablations of AGILE
> We add a detailed description of the differences of all methods from AGILE and where each of them is supposed to work (Appendix C.1 and Figure 17). To answer your questions, all ablations receive the same state and action representations as input, like AGILE. They also compute action set summaries like AGILE - using GAT, Deep Set, or LSTM. The only difference is that AGILE uses relational action features processed by the GAT, while ablations use raw action representations. AGILE is expected to outperform ablations on environments requiring many diverse action relations such that a single summary vector is insufficient to easily capture and associate each action’s decision with other actions. E.g. Because of the dynamic interactions in CREATE, each action decision depends on the availability of all other activators and tools. However, in Grid World, knowing the summary about which of the two dig-skills are available, is enough. Thus, AGILE = ablations on Grid World. We organize the results discussed in Section 6.1.3 to add this comparative discussion. Moreover, the parameters used by AGILE and ablations are similar since the hidden layer is always set to 64 for GAT, LSTM, and Deep-Set (Appendix Table 3). We also searched over the hidden layer dimension and found 64 to be the most suitable across all baselines and ablations.
>
> ### Inconclusive experimental results in RecSim
> Based on your feedback, we make two experimental revisions:
> 1. We **re-tune and experiment** with various design choices across all the baselines and our methods on RecSim. Each change is validated with experiments (Appendix B.2). Worth pointing out is the effect of changing target network sync frequency from 10 to 500. This led to improved RecSim results for all methods.
> 2. We add two more tasks to RecSim and thus test the consistency of results on  **3 separate RecSim tasks** - (i) Indirect-CPR (Original env - Figures 4-7): User Model depends on CPR but reward reflects this indirectly via stochastic clicks, (ii) Direct-CPR (Appendix B.3): User model depends on CPR and reward = click + CPR-reward. So, direct feedback. (iii) Pairing (Appendix B.4): Added a special item for each category (like activators in CREATE). Clicks can only happen when the correct pairs of items are recommended. Since action relations are diverse, AGILE is expected to outperform ablations with summary only.
>
> **Conclusion: Consistent trends with other environments on all RecSim tasks**
> - **AGILE > baselines**, always.
> - **AGILE >= ablations** and **AGILE >= AGILE-GCN**, especially when the task requires learning many action-specific relations which cannot be modeled by a single summary vector easily (e.g. pairing-RecSim and CREATE).
> - **AGILE >= AGILE-only-action**. AGILE-only-action is optimal on all RecSim tasks because action relations are predefined, based on item categories and thus do not vary when state changes. AGILE-only-action underperforms in Grid world and CREATE.
> - **Indirect v/s Direct reward signal**
> The trends across all methods are identical in both settings. This suggests that well-tuned RL methods are robust to stochastic rewards.
>
> We have revised the Section 6 result discussion to accommodate these conclusions and made appropriate changes to the Conclusion as you suggested.
>
> ### Other clarifications
> - **How relevant is the quality of raw action representations?**
> We add details of the action representations used for all environments to Appendix A. For Grid world and RecSim, we use predefined action representations. For CREATE and Real RecSys, we use learned representations. Previously, Jain et al (2020) have answered the interesting question you raised. Their Fig. 7(b) demonstrates that lower quality action representations lead to lower performance with both train and test actions using the utility-policy based architecture. Specifically, they show that encoding a set of videos of a CREATE tool behavior (high-dimensional) leads to poorer representations as compared to using state-trajectories (low dimensional). This, in turn, translates into a drop in performance.
> - **Notational Changes**
> We really appreciate your detailed comments and made the respective changes in Sec 4.1 and 4.2. These have helped improve the paper’s clarity.
> - **Details of qualitative analysis**
> We have added a better description of qualitative analysis, along with contrastive examples of why baselines fail to Sec 6.2, Fig. 6, and website.
> - **More Implementation decisions**
> Based on your feedback, we have added a discussion of implementation details to Appendix D.1 and experimentally validate the most important ones (B2.2).

---

### Author Response · Authors · 2021-11-23
**Added required details, experiments, and consistent conclusions**

We thank all the reviewers for the constructive feedback. Our key proposition that action relations are crucial for RL with varying action space was well received. We appreciate the positive comments on the problem motivation, solution insight, benchmark environments and qualitative evaluation. The main feedback was about missing additional details and inconclusive ablation analysis on the RecSim benchmark. We address each comment backed with new experiments wherever required:

### Additional details
- **Hyperparameters and network details for all methods (R1, R2, R4)**
We adopt the RL relevant hyperparameters (such as learning rate, entropy coefficient) from prior work and only tune the parameters specific to the added components (i.e. GAT) for fairness. We add Appendix details of hyperparameters tuning (D2-3) and network architectures (C.3) for all methods.
- **Discussion on the role of ablations (R1)**
All ablations receive the state and action representations as input, like AGILE. They also compute action set summaries - using GAT, Deep Set, or LSTM. The only difference is that AGILE uses relational action features processed by the GAT, while ablations use raw action representations. AGILE is expected to outperform ablations on environments requiring many diverse action relations such that a single summary vector is insufficient to easily capture and associate each action’s decision with other actions. E.g. Because of the dynamic interactions in CREATE, each action decision depends on the availability of all other activators and tools. However, in Grid World, knowing the summary about which of the two dig-skills are available, is enough. Thus, AGILE = ablations. Appendix C.1 and Figure 17 visually demonstrate this distinction.
- **Contrastive qualitative results with more details (R1, R2)**
Improved Fig. 6 shows how the baseline fails without the knowledge of available actions. It is unable to exploit the shortcut digging skills and always follows the same suboptimal path in Grid World. Likewise, it cannot identify the most common item category in RecSim, which is necessary to optimize CPR.
- **Listwise RL algorithm to deal with combinatorial action space (R3)**
We build upon the CDQN architecture (Chen et al., 2019a) which builds a list action incrementally by selecting N actions in sequence. For each list-index, a Q-network is used to select the best action. Along with the state, the currently built list is also an input to the Q-network. We add further details to Algorithm 1 and Appendix C.2.
- **Environment details such as action representations and data collection (R3)**
Added complete environment details to Appendix A. Specifically, A.2 explains how learned action representations in CREATE enable generalization to new tools (Jain et. al 2020). A.4 already details data collection and reward model training for real-data RecSys but is restructured for clarity.
- **How can domain knowledge be used? (R2)**
Added experiment to show that using domain knowledge to remove edges from the action graph improves learning speed and stability. (Appendix B.1)


### Inconclusive ablation analysis on RecSim (R1, R4)
We address this in two ways:
1. We **re-tune and experiment** with various design choices across all the baselines and our methods on RecSim. Each change is validated with experiments (Appendix B.2).
2. We evaluate on **3 separate RecSim tasks** - (i) Indirect-CPR (Original env): User Model depends on CPR but reward reflects this indirectly via stochastic clicks, (ii) Direct-CPR (Appendix B.3): User model depends on CPR and reward = click + CPR-reward. So, direct feedback. (iii) Pairing (Appendix B.4): Added a special item for each category (like activators in CREATE). Clicks can only happen when the correct pairs of items are recommended. Since action relations are complex, AGILE is expected to outperform ablations with summary only.

**Conclusions:**
- **Consistent trends with other environments on all RecSim tasks**
-- **AGILE > baselines**, always.
-- **AGILE >= ablations** and **AGILE >= AGILE-GCN**, especially when the task requires learning many action-specific relations which cannot be modeled by a single summary vector easily (e.g. pairing-RecSim and CREATE).
-- **AGILE >= AGILE-only-action**. AGILE-only-action is optimal on all RecSim tasks because action relations are predefined, based on item categories and thus do not vary when state changes. AGILE-only-action underperforms in Grid world and CREATE.
-- **Indirect v/s Direct reward signal**
The trends across all methods are identical in both settings. This suggests that well-tuned RL methods are robust to stochastic rewards.


### Other edits
We add a discussion on related work for *RL with invalid actions* (R2) and other notational (R1) and clarity edits (R3) to ease understanding.

[R1=RWVF, R2=FMiS, R3=3F2U, R4=mRLy]

---

### Decision · Program_Chairs · 2022-01-20

**Decision:**

Accept (Poster)

**Comment:**

This paper studies the problem of how to train an agent to understand relationships and dependencies among available (and potentially changing) actions in an RL environment to more efficiently solve a task. For instance, in the absence of a hammer for the task of putting up a painting on a wall, the agent could use an alternative tool like adhesive strips if available. The paper's main technical contribution is to use train a graph attention network to learn action space relationships under a given action representation. The paper demonstrates the effectiveness of this strategy on a range of environment benchmarks.

The reviewers initially brought up several lacunae in their assessment of the paper. These included the opaqueness in the explanation of the graph network structure, incremental nature of the improvement over the paper of Jain et al 2020, the lack of clear ablation studies and their message, comparisons with baselines drawn from other existing approaches potentially relevant to the setting, and the role of hyperparameters and their tuning.

In response, the author(s) provided detailed clarifications and additional experimental results. Namely, they clarified the details of the graph attention network, added ablation studies to help understand the role of this component, discussed the relevant and (in)applicability of other existing work, and supplied details about hyperparameter tuning. The author response was adequate to convince the reviewers to arrive at a consensus reflecting the positive impression of the paper.

In view of the unanimous opinion of the reviwers, I recommend acceptance of the paper.